# SINDy-RL for interpretable and efficient model-based reinforcement learning

Nicholas Zolman [1,2] ✉, Christian Lagemann [1], Urban Fasel[3], J. Nathan Kutz [4,5] & Steven L. Brunton [1]

Deep reinforcement learning (DRL) has shown significant promise for uncovering sophisticated control policies that interact in complex environments, such as stabilizing a tokamak fusion reactor or minimizing the drag force on an object in a fluid flow. However, DRL requires an abundance of training examples and may become prohibitively expensive for many applications. In addition, the reliance on deep neural networks often results in an uninterpretable, black-box policy that may be too computationally expensive to use with certain embedded systems. Recent advances in sparse dictionary learning, such as the sparse identification of nonlinear dynamics (SINDy), have shown promise for creating efficient and interpretable data-driven models in the low-data regime. In this work, we introduce SINDy-RL, a unifying framework for combining SINDy and DRL to create efficient, interpretable, and trustworthy representations of the dynamics model, reward function, and control policy. We demonstrate the effectiveness of our approaches on benchmark control environments and flow control problems, including gust mitigation on a 3D NACA 0012 airfoil at $Re = 1000$. SINDy-RL achieves comparable performance to modern DRL algorithms using significantly fewer interactions in the environment and results in an interpretable control policy orders of magnitude smaller than a DRL policy.

Much of the success of modern technology can be attributed to our ability to control dynamical systems: designing safe biomedical implants for homeostatic regulation, gimbaling rocket boosters for reusable launch vehicles, operating power plants and power grids, industrial manufacturing, among many other examples. Recently, advances in machine learning and optimization have rapidly accelerated our ability to tackle complicated data-driven tasks—particularly in the fields of computer vision[1] and natural language processing[2]. Reinforcement learning (RL) is at the intersection of both machine learning and optimal control, and the core ideas of RL date back to the infancy of both fields. An RL agent iteratively improves its control policy by interacting with an environment and receiving feedback about its performance on a task through a reward. Deep reinforcement learning (DRL) has shown particular promise for uncovering control policies in complex, high-dimensional spaces[3–10]. DRL has been used to achieve super-human performance in games[11–15] and drone racing[16], to control the plasma dynamics in a tokamak fusion reactor[17], to discover novel drugs[18], and for many applications in fluid mechanics[19–29]. However, these methods rely on neural networks and typically suffer from three major drawbacks: (1) they are infeasible to train for many applications because they require millions—or even billions[15]—of interactions with the environment; (2) they are challenging to deploy in resource-constrained environments (such as embedded devices and micro-robotic systems) due to the size of the networks and need for

[1]Department of Mechanical Engineering, University of Washington, Seattle, WA, USA. [2]Data Science and Artificial Intelligence Department, The Aerospace Corporation, El Segundo, CA, USA. [3]Department of Aeronautics, Imperial College, London, UK. [4]Department of Applied Mathematics, University of Washington, Seattle, WA, USA. [5]Department of Electrical and Computer Engineering, University of Washington, Seattle, WA, USA. ✉e-mail: nzolman@uw.edu

specialized software; and (3) they are "black-box" models that lack interpretability, making them untrustworthy to operate in safety-critical systems or high-consequence environments. In this work we improve the sample efficiency of reinforcement learning algorithms, even for high-dimensional problems, by leveraging sparse dictionary learning. Specifically, we build small, interpretable surrogate models for the environment dynamics, reward, and policy.

There has been significant research into reducing the amount of experience needed to train RL policies, such as offline RL[30], experience replay methods[31–33], transfer learning[34], and meta-learning[35,36]. Training in a low-fidelity representation of the environment is perhaps the most common way to reduce the number of interactions in a full-order environment. However, there are many cases where an analytic reduced-order model does not exist and the dynamics must be learned from data to create a *surrogate* representation of the environment. Dyna-style model-based reinforcement learning (MBRL) algorithms iteratively switch between learning and improving a surrogate model of the environment and training model-free policies inside the surrogate environment by generating "imaginary" experience[37]. Deep MBRL algorithms have shown significant promise for reducing sample complexity on benchmark environments[38] by simultaneously training neural network models of the environment. Although neural network models have recently become popular and are gaining wide adoption over traditional modeling methods, they are still overparameterized, data-inefficient, and uninterpretable.

In contrast, sparse dictionary learning provides an efficient and interpretable alternative to learn models from data, as in the sparse identification of nonlinear dynamics (SINDy)[39]. Sparse dictionary learning is a type of symbolic regression that learns a representation of a function as a sparse linear combination of pre-chosen candidate dictionary (or "library") functions. A sparse, symbolic model lends itself naturally to interpretation and analysis—especially for physical systems where the dictionary terms have physical meaning.

Importantly, SINDy has been extended to systems with control and used to design model predictive control (MPC) laws[40,41]. SINDy methods are incredibly efficient—both for model creation and deployment—making them promising for both online learning and resource-constrained control. In particular, SINDy has been used for online simultaneous dynamics discovery and optimal control[42]. Recent work used a single SINDy model to accelerate DRL and demonstrated its use on simple DRL benchmarks[43]. In this work, we generalize this DRL framework to include ensembles of dictionary models of both the dynamics and reward to accelerate learning in the low-data limit and quantify uncertainty.

## Our contributions

In this work, we develop techniques at the intersection of sparse dictionary learning and DRL for creating trustworthy, interpretable, efficient, and generalizable models that operate in the low-data limit. Building on recent advances in ensemble dictionary learning[44], we first introduce a Dyna-style MBRL algorithm that fits an ensemble of SINDy models to approximate an environment's dynamics and uses modern model-free reinforcement learning to train agents in the surrogate environments. In systems where the reward is difficult to measure directly from the observed state (e.g. limited sensor information for flow control on an aircraft), we augment the Dyna-style algorithm by learning an ensemble of sparse dictionary models to form a surrogate reward function. Finally, after training a DRL policy, we use an ensemble of dictionary models to learn a light-weight, symbolic policy, which can be readily transferred to an embedded system. Figure 1 provides a schematic of the SINDy-RL framework. We evaluate our methods on benchmark environments for continuous control from mechanical systems using the `dm_control`[45] and `gymnasium` suites[46] as well as fluid systems from HydroGym[47] and HydroGym-GPU[48]. We demonstrate that our methods can:

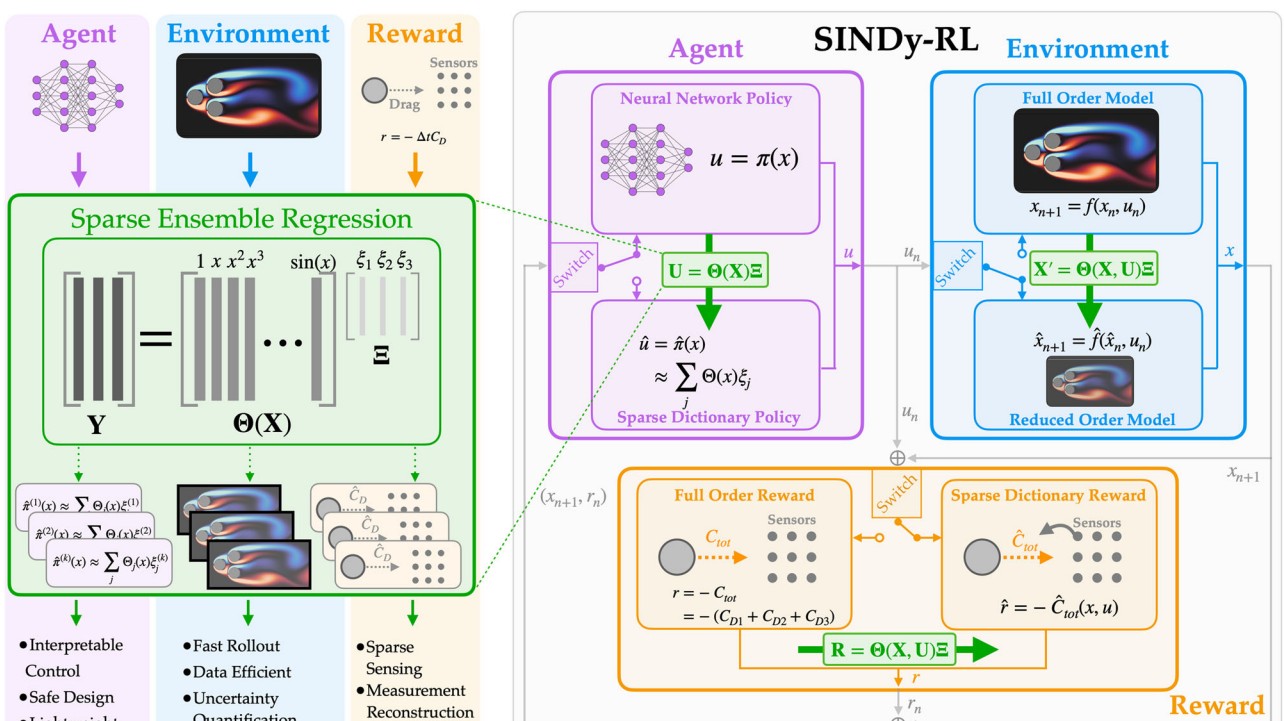

**Fig. 1 |** *Left*: SINDy-RL provides a unifying set of methods for creating efficient, interpretable models of (1) the environment dynamics, (2) the reward function, and (3) the control policy through sparse ensemble dictionary learning. Each SINDy-RL method can be used together or independently, depending on the context. *Right*: Schematic of SINDy-RL logical flow. Switches indicate which model should be used; the depicted configuration is used for training a neural network policy using a surrogate environment and reward.

- Improve sample efficiency by orders of magnitude for training a control policy by leveraging surrogate experience in an E-SINDy model of the environment.
- Leverage the efficiency of the surrogate models to accelerate expensive hyperparameter tuning.
- Learn a surrogate reward when the reward is not directly measurable from observations.
- Reduce the complexity of a neural network policy by learning a sparse, symbolic surrogate policy, with comparable performance and smoother control.
- Quantify the uncertainty of models and provide insight into the quality of the learned models.

By leveraging sparse dictionary learning in combination with deep reinforcement learning, it can become feasible to rapidly train control policies in expensive, data-constrained environments while simultaneously obtaining interpretable representations of the dynamics, reward, and control policy.

## Results

Reinforcement learning (RL) comprises a family of methods where an agent learns a policy, $\pi$, to perform a task through repeated interaction with an environment, $\mathcal{E}$. Explicitly, at each environment state $\mathbf{x}$, the agent samples an action $\mathbf{u}_n \sim \pi(\mathbf{x}_n)$ and executes it in the environment, producing a new state $\mathbf{x}_{n+1}$ and reward $r_n$—an indication of how well the agent performed at that time. We define the value function to be the expected future return for taking actions from the policy, $V_\pi(\mathbf{x}) = \mathbb{E}\left(\sum_{k=0}^{\infty} \gamma^k r_k | \mathbf{x}_0 = \mathbf{x}\right)$ where $0 < \gamma \leq 1$ is the *discount factor*. RL methods seek a policy that maximizes this quantity, which can be a very challenging optimization problem, especially in the case of high-dimensional state-spaces, continuous action spaces, and nonlinear dynamics. Deep reinforcement learning (DRL) has made significant progress in addressing these problems by parameterizing functions as deep neural networks (DNNs), such as the policy $\pi(\mathbf{x}) \approx \pi_\phi(\mathbf{x})$, and training on collected experiences.

In contrast to neural networks, dictionary learning is an efficient form of symbolic regression that models a function $\mathbf{y}(\mathbf{x}) = f(\mathbf{x})$ as a linear combination of $d$ dictionary (or "library") functions (such as polynomials, sines, cosines, etc.): $\mathbf{\Theta}(\mathbf{x}) = (\theta_1(\mathbf{x}), \theta_2(\mathbf{x}), \ldots \theta_d(\mathbf{x}))$. To learn a model from $N$ data samples of inputs $\mathbf{X} = [\mathbf{x}_1, \mathbf{x}_2, \ldots, \mathbf{x}_N]^T \in \mathbb{R}^{N \times m}$, and their associated labels $\mathbf{Y} = [\mathbf{y}_1, \mathbf{y}_2, \ldots, \mathbf{y}_N]^T \in \mathbb{R}^{N \times n}$, we evaluate the dictionary at the data, $\mathbf{X}$:

$$\mathbf{\Theta}(\mathbf{X}) = [\theta_1(\mathbf{X}), \theta_2(\mathbf{X}), \ldots, \theta_d(\mathbf{X})] \in \mathbb{R}^{N \times d}$$

to form the linear model $\mathbf{Y} = \mathbf{\Theta}(\mathbf{X})\mathbf{\Xi}$, where $\mathbf{\Xi} \in \mathbb{R}^{d \times n}$ are the coefficients to be fit. Sparse dictionary learning assumes that the desired function can be well approximated by a small subset of terms in the library, i.e. $\mathbf{\Xi}$ is a sparse matrix. For dynamics discovery, such as SINDy[39,49] (where $\mathbf{y} = \frac{d}{dt}\mathbf{x}$), a sparse library is physically motivated by the observation that the governing equations for most physical systems have relatively few terms. To achieve this parsimony, a sparse optimization problem is formulated:

$$\mathbf{\Xi}^* = \underset{\mathbf{\Xi}}{\text{argmin}} \| \mathbf{Y} - \mathbf{\Theta}(\mathbf{X})\mathbf{\Xi} \|_F^2 + \mathcal{R}(\mathbf{\Xi}) \tag{1}$$

where $\| \cdot \|_F$ is the Frobenius norm and $\mathcal{R}(\mathbf{\Xi})$ is a sparsity-promoting regularization.

Ensemble-SINDy (E-SINDy)[44] introduced a way to select a model from an ensemble of SINDy models and is more robust to noise than SINDy, particularly in the low-data limit. E-SINDy can be generalized to arbitrary dictionary models by considering ensembles: $\mathbf{Y}^{(k)} = \mathbf{\Theta}^{(k)}(\mathbf{X}^{(k)})\mathbf{\Xi}^{(k)}$, for $k = 1 \ldots N_e$. It is important to note that there have been many proposed variants of SINDy to make the algorithm more robust[50-55].

These methods are generally compatible—if not synergetic—with E-SINDy; we therefore only present the simplest formulation in this work, though a practitioner may seek to amend our framework with a more specialized variant to best suit their purpose.

In this work, we introduce SINDy-RL: a unifying set of three techniques for applying sparse dictionary learning to DRL control tasks. As shown in Fig. 1, these techniques can be used either together or separately (depending on the context) to approximate the environment dynamics, reward function, and the final learned neural network policy using ensemble sparse dictionary learning. We frequently use the "hat" notation to indicate a surrogate approximation of a function, i.e. $\hat{f}(x) \approx f(x)$. In the following, we introduce each of these new techniques and demonstrate their capability to quickly learn effective controllers for the set of five control problems depicted in Fig. 2. In particular, we demonstrate our ability to greatly reduce the amount of interaction needed with an expensive simulation environment while simultaneously extracting lightweight and interpretable representations of the dynamics, reward, and control policy.

### SINDy-RL: Learning surrogate dynamics

We propose a Dyna-style MBRL algorithm where we iteratively improve a surrogate dynamics model while learning a control policy. First, we collect offline data samples from the full-order environment by deploying a default policy (such as random input, an untrained neural network, Schroeder sweep[56], etc.). We use the collected data to fit an ensemble of SINDy models to initialize a surrogate environment. Next, we iteratively train a policy for a fixed number of policy updates using the surrogate environment with a policy optimization algorithm, such as proximal policy optimization (PPO)[57]. By forcing the agent to only interact in the surrogate environment, we offload the majority of the expensive sample collection to the lightweight E-SINDy surrogate. Finally, we deploy the trained policy to the full-order environment, collect data for evaluation, and use the newly collected data to update the E-SINDy models. We repeat this process by iteratively updating and improving both the E-SINDy model and policy. In this work, we specify a fixed number of policy iterations before evaluating the agent, though a practitioner may consider an adaptive number of updates based on training metrics in the surrogate environment. A full summary of this process can be found in Algorithm 1 under Methods.

Either a continuous or discrete model can be fit; however, we learn discrete-time models where next-step updates are predicted explicitly, rather than integrating a continuous-time model forward in time, for ease of deployment in the environment. It is common for nonlinear dynamics models—especially learned models—to grow unbounded over long time horizons unless stability guarantees are enforced[54]. To accommodate this limitation, we bound the state space and reset the surrogate environment during training if a trajectory exits the bounding box.

### SINDy-RL: Learning surrogate rewards

For Dyna-style MBRL, it is assumed that the reward function can be directly evaluated from observations of the environment. However, there are cases in which the reward function cannot be readily evaluated because the system may only be partially observable due to missing sensor information—as is the case in many fluids systems. There has been significant work on creating proxy rewards for controls tasks using "reward shaping"[58] and learning an objective function through inverse reinforcement learning[59]. We propose learning a proxy reward, $\hat{R}(\mathbf{x}_{k+1}, \mathbf{u}_k)$, with supervised sparse dictionary learning when there are offline evaluations of the reward available: $r_k = R(\mathbf{x}_{k+1}, \mathbf{u}_k)$. Our implementation uses sparse ensemble dictionary learning under the assumption that there is a sparse deterministic relationship between the reward function and observations from the environment. We incorporate this into Algorithm 1 by learning the reward function alongside the dynamics to create the surrogate environment.

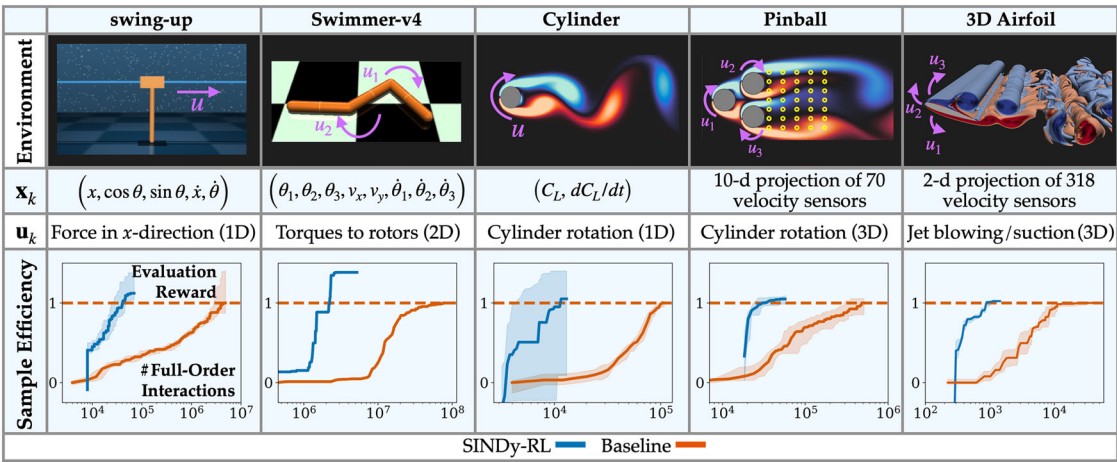

**Fig. 2 | Overview of environments and results.** *Environment.* The five environments used and their corresponding observations, $\mathbf{x}_k$, and actions $\mathbf{u}_k$: (1) `swing-up` attempts to swing a pole and stabilize it in the upright position; (2) `Swimmer-v4` controls a 3-segment robot to travel as far as possible in the horizontal direction (along the $x$-axis) in a fixed amount of time; (3) Cylinder reduces the drag force, $C_D$, of a rotating cylinder in an unsteady fluid flow with measurements of the lift force at $Re = 100$; (4) Pinball reduces the net drag force $C_{D1} + C_{D2} + C_{D3}$ on the system of cylinders to stabilize the quasi-periodic flow at $Re = 100$; and (5) 3D Airfoil mitigates the effect of a large incoming gust in an unsteady flow at $Re = 1000$ by minimizing $|C_L - C_L^{ref}| + 0.25|C_D - C_D^{ref}|$, i.e. deviations of the lift force, $C_L$, and drag force, $C_D$, from reference values. For Pinball and 3D Airfoil, the observations are produced by projecting sensor measurements onto leading SVD modes from data. For Pinball, the yellow circles indicate the location of the sensors. A detailed schematic of the 3D Airfoil can be found in Fig. 4. More details about each environment can be found under Methods and Supplementary Section 3. *Sample Efficiency:* SINDy-RL sample efficiency comparison with a Baseline DRL approach, PPO (including initial offline data collection detailed in Supplementary Table 2). Shaded regions indicate the performance for the 25th to 75th quantiles among independent agents. Evaluation rewards are scaled based on the median baseline performance. `Swimmer-v4` comparison uses PBT with top-performing agent.

## SINDy-RL: Learning surrogate policies

Taking inspiration from behavior cloning algorithms in imitation learning[60], we fit a sparse dictionary model approximation of the final neural network learned policy, $\pi_\phi$. While imitation learning traditionally attempts to mimic the policy of an expert actor with a neural network student, we instead use the learned neural network as our expert and the dictionary model as our student, resulting in a lightweight, symbolic approximation. A dictionary model is not as expressive as a neural network; however, there has been previous work indicating that even limiting to purely linear policies can be a competitive alternative to deep policy networks[61,62], showing that even complicated control tasks may have a simple controller. Likewise, there has been recent investigation into approximating neural networks with polynomials through Taylor expansion[63], which has shown to provide sufficiently robust approximations. These reduced representations can be orders of magnitude smaller than a fully-connected neural network and they can be efficiently implemented and deployed to resource-constrained environments, such as embedded systems. Many model-free algorithms assume a stochastic policy for optimally interacting with a stochastic environment and encouraging exploration; we proceed with a sparse ensemble fit of the expectation: $\hat{\pi}(\mathbf{x}) \approx \mathbb{E}[\pi_\phi(\mathbf{x})]$. Because there is no temporal dependence on $\pi_\phi$, we can assemble our data and label pairs $(\mathbf{x}, \mathbb{E}[\pi_\phi(\mathbf{x})])$ by evaluating $\pi_\phi$ for any $\mathbf{x}$.

## Using SINDy to accelerate DRL training

By quickly developing a SINDy surrogate model using limited data, it is possible to offload the expensive interaction with a full-order model. In Fig. 2, we summarize how SINDy-RL reduces the number of interactions with our five evaluation environments compared to the popular model-free proximal policy optimization (PPO) algorithm[57]; in each case, SINDy-RL improves the sample efficiency by $10 - 100 \times$. In Fig. 3, we also provide a more in-depth comparison our proposed SINDy-RL method from Algorithm 1 with a quadratic dynamics library to four different baseline experiments: (1) PPO, (2) Algorithm 1 with a linear SINDy library, (3) Algorithm 1 where SINDy models are replaced with neural network dynamics models (i.e. "Dyna-NN"), and (4) RLlib's

implementation of Model-Based Meta-Policy Optimization (MB-MPO)[64]. As shown in Fig. 3, SINDy-RL with a quadratic library learns a control policy with $100 \times$ fewer interactions in the full-order environment compared to model-free RL and greatly outperforms other model-based approaches. The linear models are incapable of approximating the global dynamics since there are multiple fixed points. In contrast, while neural network models should be more expressive, it appears that both the MB-MPO and Dyna-NN baselines fail to accurately capture the dynamics sufficient to achieve the task. For both comparisons, it is suspected that the dynamics may be captured with significantly larger data collections—as shown in previous work comparing SINDy with neural network models for MPC[40]. An investigation of the hyperparameters from our method can be found in Supplementary Section 4A.

The increased sample efficiency is particularly important for applications interacting with physical systems that require humans in-the-loop to monitor and reset environments or computationally expensive simulations—such as CFD solvers—that require significant time and resources to run. The benefits of increasing the sample efficiency can be seen using the fluid flow control environments. The 3D Airfoil environment (Fig. 4) uses the lattice Boltzmann method with over 72M cells, requiring 75 GB of VRAM across four A100 GPUs. Even when using the GPUs, a single step in the full-order model takes on the order of 45s to update the environment. In comparison, the learned E-SINDy polynomials for the 3D Airfoil require less than 10kB of RAM on CPU, and a single step takes approximately 1-4 milliseconds using NumPy[65]—i.e., SINDy-RL experience collection is $10^4 \times$ faster than the full-order CFD models. The SINDy-RL agent was $14.47 \times$ more sample efficient for the 3D Airfoil environment; with only 25 dynamics updates (i.e. iterations of Step 2 from Algorithm 1), the median SINDy-RL agent reached the final median baseline agent performance. Physically, this was the difference between 14 hours of training using the SINDy-RL environment and 185 hours using the full-order environment, greatly reducing the total amount of time needed to learn capable policies. A comparison of clock times for different aspects of training the HydroGym environments can be found in Supplementary Table 4.

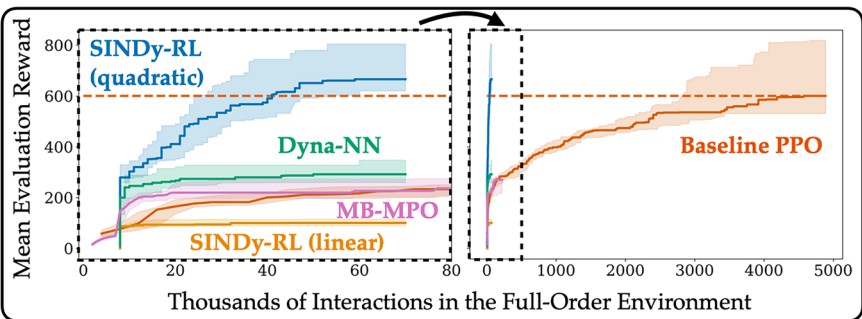

**Fig. 3 | Swing-up Sample Efficiency.** Comparison of the number of interactions in the full-order `swing-up` environment for different algorithms. For each algorithm, twenty independent seeds were run and periodically evaluated on 5 randomly generated episodes. Each agent's best average performance was tracked, updated, and recorded. The shaded area corresponds to the 25th and 75th quantiles, and the bold lines are the median. The dashed red line corresponds to the median best performance of the baseline PPO agent before training was stopped.

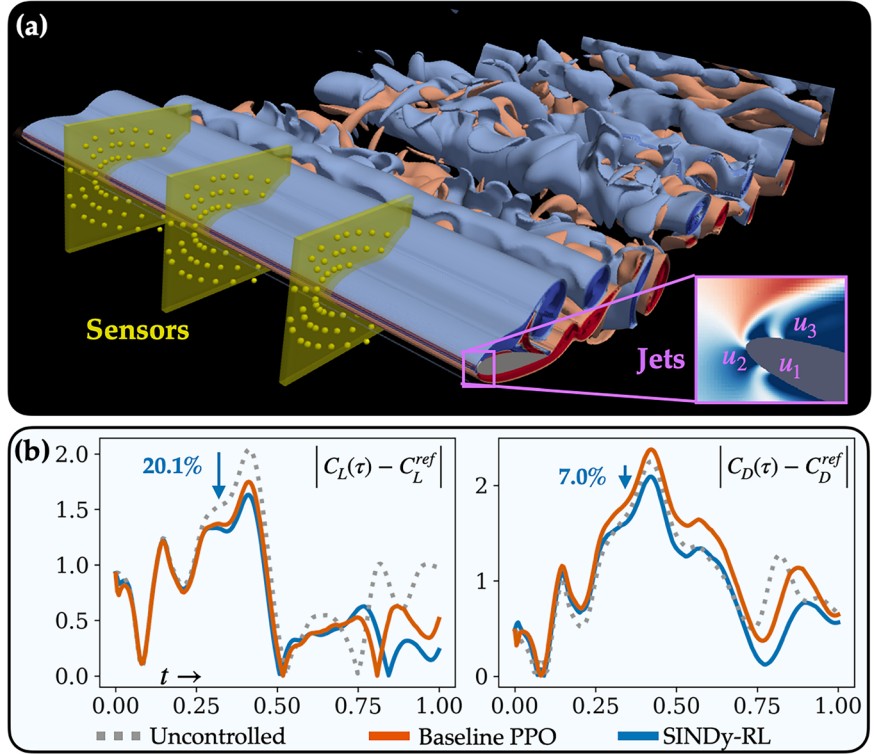

**Fig. 4 | 3D Airfoil. a** Schematic of the 3D Airfoil environment at $Re = 1000$, 20° angle-of-attack, and $M = 0.2$. The gust profile follows a 1-cosine approach with a gust factor $G = 2.0$. Q-criterion isosurfaces of the vorticity demonstrate the presence of 3D instabilities. Sensors are restricted to three planes, each consisting of 53 2-d velocity probes. Actuation is performed using three independent jets along the spanwise direction. **b** Effect of SINDy-RL policy on the normalized lift and drag forces used in the reward function. Compared to the uncontrolled environment, SINDy-RL reduces the peak error from the reference lift, $C_L$, and drag, $C_D$, values by 20.1% and 7.0% respectively. The best trained baseline PPO agent was able to reduce the peak $C_L$ error by 14.3%, but rose the peak $C_D$ error by 5.7%.

It is important to note the large variance in performance among the 20 trained SINDy-RL Cylinder agents depicted in the shaded region of Fig. 2. After interacting with the full-order Cylinder environment 13,000 times, the top 50% performing SINDy-RL agents were able to surpass the best baseline performance—some achieving 11% drag reduction, comparable with optimized solutions from previous literature[66,67]. However, the bottom 25% of agents performed poorly. Surrogate dynamics models can quickly diverge—especially early in training—which can provide misleading rewards. When these effects are further coupled with unfavorable policy network initializations, agent performance can severely degrade. The overall success of the majority of the agents indicates that methods like *population-based training* may be extremely effective at pruning these bad combinations

of surrogate dynamics, rewards, and policies to quickly find exceptional policies using very few interactions in the full-order environment.

### Accelerating hyperparameter tuning

The success of DRL to find effective control policies can often depend on the neural network initialization and choice of hyperparameters; otherwise training might lead to a suboptimal policy. To address this, it is common to use different random initializations for the network and hyperparameter tuning, and software packages have been created to facilitate these searches, such as Ray Tune[68]. While these tuning algorithms can be effective for uncovering sophisticated policies, they generally rely on parallelizing training and significant compute usage.

Previous benchmarks[69,70] have shown that the `Swimmer-v4` environment is particularly challenging to train with PPO and it has been suggested[71] that learned policies for `Swimmer-v4` are especially sensitive to certain hyperparameters and tuning can achieve superior performance. We pursue this idea by comparing population-based training (PBT)[72] to improve policies for agents interacting in the full-order `Swimmer-v4` environment and a SINDy-RL environment with dictionary surrogates for the dynamics and reward. Both experiments used a population of 20 policies and periodically evaluated the policies using the rollouts from the full-order model. As shown in Fig. 2, the SINDy-RL training is able to achieve nearly 30% better maximal performance than the baseline using 50 × fewer samples from the full-order environment. This indicates that SINDy-RL can provide a convenient way to accelerate hyperparameter tuning in more sophisticated environments. Additional training details for PBT can be found in Supplementary Section 4B.

### Generalizing to unseen dynamics

It has been well-documented that while DNNs are excellent at performing on unseen data sampled from the same data distribution used for training, i.e. interpolation, they often fail at extrapolation to data beyond the convex hull of the training set. In Fig. 5, we investigate the ability of top-performing Pinball agents trained at $Re = 100$ for 20-second segments to extrapolate to unseen dynamics at $Re = 150, 250$, and 350 for a 100-second evaluation. It is well-known that the dynamics of the Pinball system undergo a bifurcation at around $Re = 115$ where the dynamics transition from being quasi-periodic to chaotic[73]; thus, the dynamics are fundamentally different at these $Re$ values, which can be qualitatively seen—especially at $Re = 350$—in the initial conditions shown in Fig. 5(b). Despite these fundamental differences, the SINDy-RL policy is able to reasonably extrapolate. While performance does degrade compared to $Re = 100$, the agent is able to

obtain a much better cumulative return of reward over the long trajectory compared to the baseline neural network counterpart.

### Interpretable surrogate dynamics

Despite none of the evaluation environments having a polynomial representation of the dynamics, we find that surrogate models with polynomial dynamics often provide a sufficient representation of the environment to learn a control policy through repeated interaction. A thorough investigation of the learned dynamics from each environment can be found in Supplementary Section 5.

We find that for most systems, it is imperative to periodically refit the dynamics while the agent explores control strategies. For example, with the `swing-up` dynamics, the agent has no information about the goal state at the unstable equilibrium early in training. In Supplementary Fig. 13, we show that the first learned dynamics model provides a reasonable representation of the phase portrait, but the learned unstable fixed point is offset from the ground-truth position. Without refitting the dynamics, the agent would learn to drive the system to the wrong point and never stabilize the true system. However, by deploying the learned policy on the full-order system and gathering new data, the quality of the dynamics model improves and the agent is able to complete the task both using the surrogate and full-order dynamics. Finally, unlike a neural network representation of the dynamics, our approach provides a symbolic representation of the learned dynamics. In Supplementary Section 5, we provide the learned coefficients $\Xi$ and show that the `swing-up` dynamics model is well-represented by an Eulerian integration scheme: $\mathbf{x}_{k+1} = \mathbf{x}_k + \Delta t f(\mathbf{x}_k)$.

It is important to note a limitation of using dictionary dynamics; for large observation spaces, the size of a polynomial dictionary increases combinatorially. Therefore for large spaces, dimensionality reduction is critical. We demonstrate the viability of this approach for both the Pinball and 3D Airfoil environments by linearly projecting the

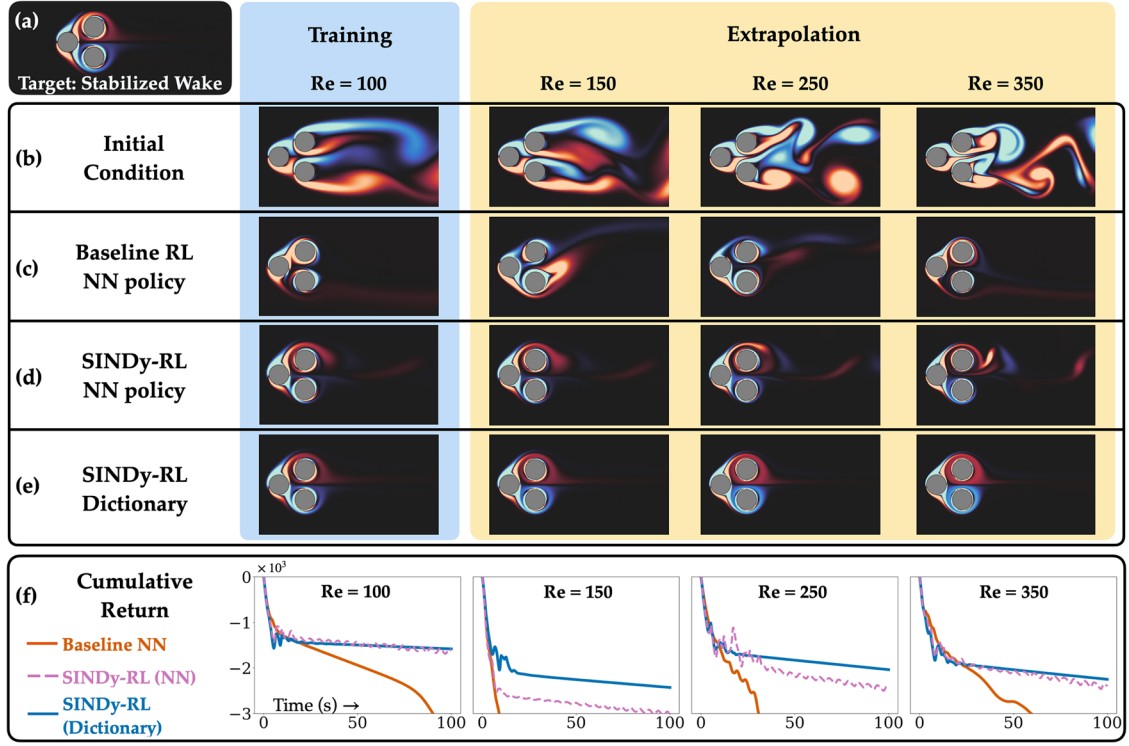

**Fig. 5 | Evaluation of Pinball polices for various *Re*. a** Vorticity snapshot of the target state with the wake stabilized. **b** Vorticity snapshots of initial conditions for the system at each *Re*. **c**–**e** evaluation of the baseline PPO, SINDy-RL (with NN policy), and dictionary policy distilled from from the SINDy-RL NN. Policies were only trained using *Re* = 100. Each snapshot is taken after 100s of feedback control. **f** Comparison of the cumulative return over time for the policies across different *Re*; accumulation of negative rewards leads to a decreasing curve.

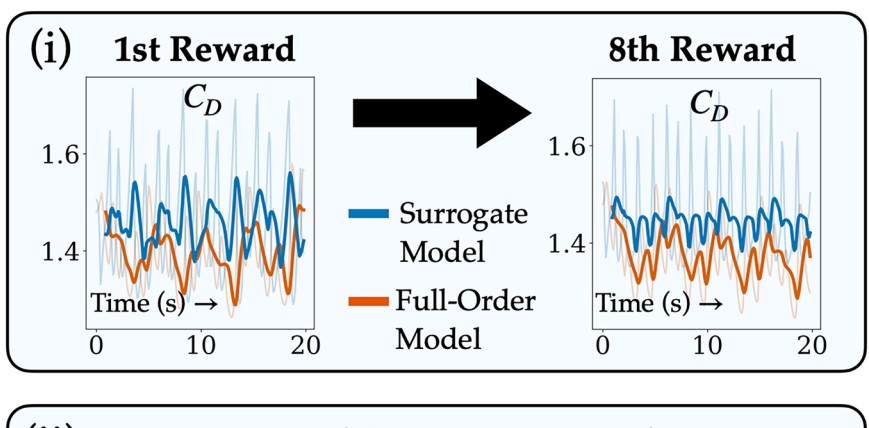

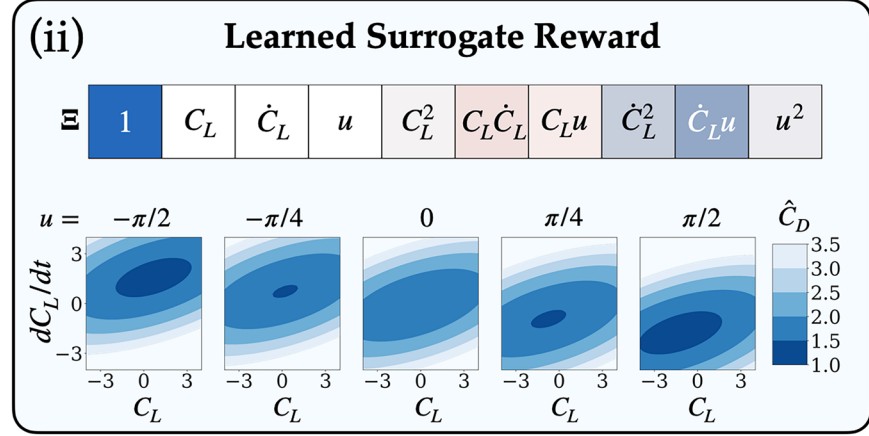

**Fig. 6 | Cylinder Reward.** (i): Comparison of the learned surrogate reward $\hat{r}(C_L, dC_L/dt, u) = -\hat{C}_D \Delta t$ for the Cylinder environment at the beginning of training and after 8 dynamics and reward updates. (ii, *Top*): a heatmap of the median dictionary coefficients $\Xi$, (ii, *Bottom*): a series of contour plots show the effect of control on $\hat{C}_D = -\hat{r}/\Delta t$.

70- and 318-dimensional observations onto the dominant SVD modes of the system and found this to be sufficient for purposes of training. For more complicated dynamics, autoencoders have been found to be useful for finding nonlinear projections[74].

**Interpretable reward functions**

In the setting where exact rewards are not analytically expressible from observations (e.g., all of our evaluation environments except swing-up), we periodically fit the surrogate dictionary reward $\hat{r} = \hat{R}(\mathbf{x}, \mathbf{u})$ alongside and independent of the SINDy dynamics. For Swimmer-v4, the rewards in the full-order environment are given by the agent's body-centered horizontal velocity. However, this quantity is not provided by the observation space. It has been well-documented that—even with access to the exact rewards—solving the Swimmer-v4 task is a considerable challenge because of the placement of the velocity sensors; researchers have even created their own modified versions of the environment when performing benchmarks[38]. Our method identified that a sparse reward of $\hat{r} \approx v_x$—the horizontal velocity of the leading segment (not the body)—was a suitable surrogate and remained stable throughout the entirety of the population-based training.

In contrast to the Swimmer-v4 environment, the Cylinder, Pinball, and 3D Airfoil environments are governed by nonlinear PDEs, where the rewards are scalar measurements evolving with the dynamics on the domain and are not analytically expressible in terms of the provided observations. The lack of available information makes modeling the environment and the reward a very challenging task. Just like the dynamics, Fig. 6(i) demonstrates the necessity of periodically updating the surrogate models. At the beginning of training, the learned reward is actually anti-correlated with the full-order reward; however, the learned reward becomes well-correlated after subsequent updates. Despite the inability to learn the exact reward in the fluid environments, the surrogate reward provides a sufficient enough learning

signal for training a comparable control policy when having access to the exact rewards. We also highlight the interpretability of our method in Fig. 6(ii); the learned reward function is a quadratic polynomial of the observations, and we can analyze the explicit influence of the control. The bowl-shaped function (given by the quadratic) gradually shifts in response to the increasingly positive control. Furthermore, it is clear that the drag is minimized (i.e., reward is maximized) for large values of $|u|$, indicating that an optimal control strategy would apply maximal control input to stay in the bowl's minimum. For the Pinball and 3D Airfoil environments, the sparse sensors in the flow are not sufficient to describe the net forces acting on surfaces; indeed, the discovered reward functions are dominated primarily by terms coupling the actuation (local information) to the sensors. The learned rewards and a detailed investigation for each environment can be found in Supplementary Section 6.

**Interpretable surrogate policies**

While DNN policies can find solutions to complicated control problems, they are typically over-parameterized, black-box models that lack interpretability. There have been several approaches to combine symbolic regression and DNNs to improve interpretability[55,75–79], but they tend to focus on discovering the dynamics rather than a symbolic form for a controller. In this work, we discover lightweight models of the control policies with sparse dictionary learning using behavior cloning; for each environment, we leverage the E-SINDy dynamics and the neural network policy obtained from Algorithm 1 to collect data for fitting the dictionary model.

Figure 5 compares the performance of the SINDy-RL neural network policy to the sparse dictionary policy distilled from it in the Pinball environment. As shown in Fig. 5e, the dictionary approach is able to consistently stabilize the wake across various values of $Re$ compared to its neural network counterpart. This is shown

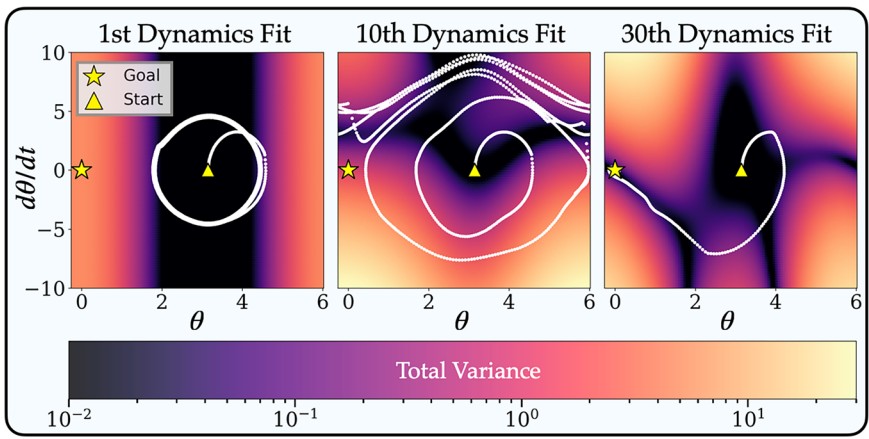

**Fig. 7 | Swing-up Dynamics Variance.** The variance of learned dynamics across snapshots during SINDy-RL training. For ease of visualization, the landscape is evaluated at $x = dx/dt = u = 0$.

quantitatively in Fig. 5f where the dictionary policy consistently receives higher cumulative returns during long evaluations of the policy. The final dictionary control policy is represented as a quadratic polynomial library containing 66 elements, nearly two orders of magnitude fewer parameters than the corresponding neural network policy with over 5000 parameters. This is also considered a relatively small neural network model; for deeper or wider networks, one can reasonably expect even larger gains. In Supplementary Section 7B, we provide further evidence that a dictionary policy can produce smoother and more consistent control inputs—outperforming the neural network when it has overfit to a particular trajectory in the `swing-up` environment.

However, the Cylinder environment provides an example where this method may struggle; the surrogate policy only reduces drag by about 3.7% compared to the neural network policy's 8.7% reduction. The original neural network agent learned a bang-bang control policy; this is very difficult to approximate with a smooth polynomial due to the bounded derivatives. One of the key challenges that supervised imitation learning faces (e.g., behavior cloning) is the issue of compounding errors during policy deployment. Because our policy approximation is a type of behavior cloning, our method inherits this challenge; the approximate policy slowly drifts away from the state-action pairs it was trained on and ultimately performs suboptimally. A detailed investigation of the learned dictionary policies for each environment (including a comparison of different data sampling strategies used for the behavior cloning) can be found in Supplementary Section 7.

### Uncertainty quantification

Treating a set of dictionary coefficients, $\Xi$, as random variables, an ensemble acts as an empirical approximation for the distribution of likely $\Xi$ values. From this, one can derive an efficient framework for analytically approximating a model's pointwise variance. Explicitly, for $\Xi = [\xi_1, \dots \xi_n]$, and letting $\text{Cov}(\xi_i)$ denote the sample covariance matrix of $\xi_i$ computed from the ensemble, the uncertainty of $\mathbf{y}$ given $\mathbf{x}$ is:

$$\text{Tr}\big(\text{Var}_\Xi[\mathbf{y}(\mathbf{x})|\mathbf{x}]\big) = \sum_{i=1}^{n} \Theta(\mathbf{x})\text{Cov}(\xi_i)\Theta(\mathbf{x})^T. \tag{2}$$

In Fig. 7, we visualize the evolution of the uncertainty landscape for the `swing-up` dynamics when training the dynamics and policy with SINDy-RL. At the beginning of training, the dynamics are confined to a narrow band of low-variance states where data has been previously collected. During training, the variance landscape changes in response to new, on-policy data. By the time the agent has learned to solve the `swing-up` task, the learned dynamics have become more certain and the agent takes trajectories over regions that have less variance.

In this work, we only use variance to inspect the learned dictionary functions, although there are further opportunities to exploit the variance. There is a trade-off in sample efficiency between refining the learned dynamics and improving the policy early in training. With a poor representation of the surrogate dynamics, we risk overfitting to the surrogate and discovering policies that do not generalize well to the real environment. Analogous to curiosity-driven learning[80], the estimated uncertainty of the dynamics and reward models may strategically guide the exploration of the environment and rapidly improve them—making it less necessary to query the environment later. Likewise, we may encourage "risk-averse" agents by penalizing regions with high uncertainty—encouraging agents to take trajectories where the dynamics are trusted and agree with the full-order model.

## Discussion

This work developed a unifying framework for combining SINDy (i.e., sparse dictionary learning) with deep reinforcement learning to learn efficient, interpretable, and trustworthy representations of the environment dynamics, the reward function, and the control policy, using significantly fewer interactions with the full environment. We demonstrate the effectiveness of SINDy-RL on several challenging benchmark control environments, including performing gust mitigation of NACA 0012 airfoil at $Re = 1000$ in a 3D, unsteady environment.

By learning a sparse representation of the dynamics, we developed a Dyna-style MBRL algorithm that could be $10 - 100 \times$ more sample efficient than a model-free approach, while maintaining a significantly smaller model representation than a black-box neural network model. When the reward function for an objective is not easily measurable from the observations—e.g., with only access to sparse sensor data—SINDy-RL can simultaneously learn dictionary models of the reward and dynamics from the environment for sample-efficient DRL. We also demonstrated that SINDy-RL can learn a sparse dictionary representation of the control policy for certain tasks; the resulting sparse policy is (1) orders of magnitude smaller than the original neural network policy, (2) has a smoother structure, and (3) is inherently more interpretable. With a lightweight polynomial representation of the control policy, it becomes more feasible to transition to embedded systems and resource-limited applications. The interpretable representation of the dictionary policy also facilitates classical sensitivity analysis of the dynamics and control, such as quantifying stability regions and providing robust bounds—an especially important quality in high-consequence and safety-critical environments. Finally, we have shown that for dictionary models, it is possible to analytically compute the point-wise variance of the model to efficiently estimate the uncertainty from an ensemble, which can provide insight

into the trustworthiness of the model and possibly be used for active learning by intelligently steering the system into areas of high-uncertainty.

A key ingredient for successfully applying DRL is to learn over long time-horizons. This posed a significant challenge to SINDy-RL (and Dyna-style learning more broadly) because the learned dynamics models are not guaranteed to be stable or converge—especially under the presence of control. We address this by incorporating known constraints, such as resetting the environment if a predicted state value exits a bounding box and projecting the state-space back onto the appropriate manifold after each step. There has been substantial work constraining dictionary dynamics models with structured priors—such as conservation laws[81], symmetry[82], stability regions[54], and other forms of domain knowledge[83,84]—which further offer many promising avenues for practitioners to encourage agents to stably interact over long time horizons.

Due to the combinatorial scaling of the library, dictionary learning is challenging to apply directly to high-dimensional spaces. In this work, we have demonstrated that projecting onto a low-dimensional linear subspace using the SVD can be sufficient to make this tractable. Subsequently, SINDy-RL was applied to controlling PDEs[85] by using the SINDy autoencoder framework[74] to discover nonlinear projections of the observation and action spaces onto a low-dimensional manifold. Discovering a coordinate system where the dynamics are smooth and globally defined may also help address the challenges that dictionary approaches face when the model discovery is piecewise or discontinuous. Partial observability of the environment also poses challenges for this framework. We have shown that sometimes there is sufficient information available to model the reward function from observations, but in practice, this will not always be the case. Recent work[86] has investigated deep delay embeddings to identify governing dynamics when there is substantial missing information, which is an opportunity for future investigation.

While we have restricted our attention in this work to using SINDy for Dyna-style MBRL by exploiting a model-free DRL algorithm, it is important to note that there are further opportunities to combine SINDy with control. Instead of DRL, a gradient-free policy optimization, such as evolutionary algorithms[87], could completely replace a model-free DRL optimizer. Furthermore, SINDy provides an analytic, differentiable representation of the dynamics, thus SINDy can act as a differentiable physics engine for control and be used for directly calculating gradients of RL objectives. For example, by modeling the dynamics and value functions as dictionary models, the differentiable structure was utilized with the Hamilton-Jacobi-Bellman equations to directly calculate the optimal control of a system[42]. Finally, there may be a way to bypass the use of a policy network altogether by representing the policy directly as a sparse dictionary model for policy gradient optimization.

## Methods
### Learning surrogate dynamics
Algorithm 1 gives an overview of the Dyna-style MBRL algorithm. Each dynamics model in the ensemble is fit using the discrete form of SINDy with control (SINDy-C)[40] and STRidge[49]. Hyperparameters used to learn the dynamics for each environment can be found in Supplementary Section 5.

### Algorithm 1. (Dyna-style SINDy-RL)
Input: $\mathcal{E}$ ▷ full-order environment
 $N_{\text{off}}, N_{\text{collect}}$ ▷ # of off- and on-line policy steps
 $n_{\text{batch}}$ ▷ # of policy iters / SINDy update
 $\Theta$ ▷ dictionary functions
 $\mathcal{A}$ ▷ policy optimization algorithm
 $\pi_0$ ▷ default policy
Step 1: **Initialize Surrogate Environment**
 $\mathcal{D}_{\text{off}} = \text{CollectData}(\mathcal{E}, \pi_0, N_{\text{off}})$
 $\mathcal{D} = \text{InitializeDatastore}(\mathcal{D}_{\text{off}})$
 $\Xi = \text{ESINDy}(\mathcal{D}, \Theta)$
 $\hat{\mathcal{E}} = \text{Surrogate}(\Xi)$
Step 2: **Model Improvement**
 $\pi = \text{InitializePolicy}()$
 **while** not done: **do**
 $\pi = \mathcal{A}(\hat{\mathcal{E}}, \pi, n_{\text{batch}})$
 $\mathcal{D}_{\text{on}} = \text{CollectData}(\mathcal{E}, \pi, N_{\text{collect}})$
 $\mathcal{D} = \text{UpdateStore}(\mathcal{D}, \mathcal{D}_{\text{on}})$
 $\Xi = \text{ESINDy}(\mathcal{D}, \Theta)$
 $\hat{\mathcal{E}} = \text{Surrogate}(\Xi)$
 **end while**
Output: Optimized policy, $\pi$; SINDy environment, $\hat{\mathcal{E}}$.

### Learning surrogate policies
To build our dataset, we sample points from trajectories of the agent interacting in the environment, ensuring that the states are directly relevant to the controls task. To avoid the cost of collecting more data from the full-order environment, $\mathcal{E}$, we propose *sampling new trajectories from a learned ensemble of dynamics models*. Explicitly, we sample $N_\tau$ trajectories by propagating the E-SINDy model

$$\tau^{(i)} = \{(\mathbf{x}_k^{(i)}, \mathbf{u}_k^{(i)}, \mathbf{x}_{k+1}^{(i)}, r_k^{(i)})\}_{k=1}^{T_i}, \qquad i = 1, \dots N_\tau,$$
$$\mathbf{u}_k = \mathbb{E}[\pi_\phi(\mathbf{x}_k)], \qquad\qquad \mathbf{x}_{k+1} = \hat{f}(\mathbf{x}_k, \mathbf{u}_k)$$

and use the collected data $(\mathbf{x}_k, \mathbf{u}_k)$ from each $\tau$ to fit the dynamics model. During the DRL training, the neural network policy may overfit to specific regions of the space and bias our data collection; thus, we draw inspiration from tube MPC[88], where an MPC controller attempts to stay within some bounded region of a nominal trajectory. Instead of only sampling points from the surrogate trajectories, we can sample plausible points in a neighborhood of the trajectories to create a more accurate approximation while still avoiding regions of the space that can no longer be trusted.

### Evaluation environments
We evaluate each our SINDy-RL techniques on five environments depicted in Fig. 2: (1) `dm_controlswing-up`[45] balances a pole on a cart in the unstable upright position starting from the stable down position at rest, (2) `gymnasiumSwimmer-v4`[46] controls a 3-segment robot to travel as far as possible in the horizontal direction (along the $x$-axis) for a fixed time, (3) HydroGym Cylinder[47] reduces the drag force, $C_D$, of a rotating cylinder in an unsteady fluid flow at $Re = 100$ with measurements of the lift force, $C_L$, (4) HydroGym Pinball reduces the net drag force $C_{D1} + C_{D2} + C_{D3}$ on the system of cylinders to stabilize the quasi-periodic flow at $Re = 100$, and (5) HydroGym-GPU 3D Airfoil[48] mitigates the effect of a large incoming gust in an unsteady flow at $Re = 1000$ by minimizing $|C_L - C_L^{ref}| + \frac{1}{4}|C_D - C_D^{ref}|$, i.e. deviations of $C_L$ and $C_D$ from reference values. For Pinball and 3D Airfoil, state information is obtained from a mesh of velocity probes in the flow, forming a 70- and 318-dimensional measurement space, respectively. This data is projected onto the leading singular value decomposition modes (SVD) and uses the coefficients, $a_i = \mathbf{v}_i^T \mathbf{x}$, to form a 10- and 2-dimensional observation space for the respective systems. Details for each environment can be found in Supplementary Section 3. Whereas the `swing-up` reward is analytically expressible in terms of the observed variables, all other rewards are computed from environment information that cannot retrieved analytically from the observation. We treat the rewards as functions of the observations and actions by approximating them with an ensemble of sparse models.

### Benchmarking
PPO is used as the policy optimization algorithm, $\mathcal{A}$, for all Dyna-style experiments with identical hyperparameters between comparisons. For every algorithm and experiment, multiple independent

instantiations are run to provide a distribution of performance across random seeds. Twenty instantiations were used for all environments, except 3D Airfoil where we are limited to four instantiations due to the computational demand of the environment. Specific details about the training, hyperparameters, and more can be found in Supplementary Section 4.

## Materials

All experiments, with the exception of the 3D Airfoil environment, were performed using a single-node, Linux engineering workstation consisting of a total of 40 CPUs (Intel® Xeon® Gold 6230). The 3D Airfoil experiments used NVIDIA A100 GPUs on JUWELS Booster and JURECA at the Jülich Supercomputing Centre (JSC) / Forschungszentrum Jülich.

## Data availability

Data from trained agents are publicly available and available in the HuggingFace database under accession code https://doi.org/10.57967/hf/[89]https://huggingface.co/nzolman/sindy-rl_data.

## Code availability

Code and training configurations are publicly available in Zenodo under accession codes https://doi.org/10.5281/zenodo.17087492[90] and https://doi.org/10.5281/zenodo.17088004[91]. Code is also hosted in our GitHub repository at https://github.com/nzolman/sindy-rl.

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

## Acknowledgements

The authors acknowledge support from the National Science Foundation AI Institute in Dynamic Systems grant number 2112085 (NZ, CL, JNK, SLB). SLB acknowledges support from the Army Research Office (W911NF-19-1-0045) and the Boeing Company. NZ acknowledges support from The Aerospace Corporation. CL acknowledges support from the German Research Foundation within the Walter Benjamin fellowships LA 5508/1-1. The authors gratefully acknowledge the Gauss Centre for Supercomputing e.V. for funding this project by providing computing time on the GCS Supercomputers. Furthermore, the authors would like to thank the HydroGym developers—especially Samuel Ahnert—for their help and input regarding the HydroGym examples. Finally, the authors would like to acknowledge the helpful feedback from the anonymous reviewers in strengthening the paper.

## Author contributions

N.Z. designed, performed research, and analyzed results; all authors were involved in discussions to interpret results; C.L. designed the 3D Airfoil simulation environment and analyzed its results; N.Z. and C.L. implemented the methods on the different simulation environments; U.F. helped design and formulate the methods; S.L.B. and J.N.K. received funds to support this work; N.Z. wrote the paper, and all authors helped to review and edit.

## Competing interests

The authors declare no competing interests.
