## [Transparent Peer Review file · Nature Communications]

SINDy-RL: Interpretable and Efficient Model-Based Reinforcement Learning

Corresponding Author: Mr Nicholas Zolman

Version 0:

Reviewer comments:

Reviewer #1

(Remarks to the Author)

Please find the attached file.

(Remarks on code availability)

Reviewer #2

(Remarks to the Author)

This paper combines advances in physics and machine learning by introducing SINDy-RL---a method that combines sparse learning of dynamics and reinforcement learning. SINDy-RL is a Dyna-style algorithm for Model-Based Reinforcement Learning which fits the dynamics model of the world, the reward function, and the policies.

This work is novel as compared to past work as it also learns dynamics models for the reward function and policies. Learning the reward function is useful in complex systems, and learning a representation of the policies can help in time-sensitive applications where large neural networks are slow (although learning a linearly-combination of non-linear terms to express a policy can cause in loss of accuracy).

This paper has merit since the authors compare their learned policies against the ground truth dynamics functions---providing evidence that on the working of the method.

I believe the paper can be improved with the following tweaks:

1. Currently, the environments being tested against are low-dimensional. Particularly, pendulum-swing-up and swimmer are lower dimensional environments.

These environments have their merits since their lower-order allows for easier explainability.

However, results on a higher-order environment will solidify my trust in the methods working. Examples could include Hopper, Halfcheetah, or more.

I recommend adding these results in.

2. Refitting to SINDy with large amounts of data can be expensive. While the model itself may not be computationally intensive as compared to a neural network, the training process could take longer.

A plot that compares the reward vs the clock time would be beneficial. For instance, it can take vanilla PPO 3 hours to reach an optimal reward on a task, and it can take SINDy-RL 6 hours to do the same---while SINDy-RL took less interactions with the environment, vanilla PPO took lesser clock time.

In environments where it is very difficult to interact with the environment, such as robotics, I would choose SINDy-RL. In cases where interactions with the environment are not costly, I would choose the neural network and then prune it.

Having a clock time comparison makes the argument stronger. Knowing more about the time it takes to fit a SINDy model when adding additional data on-policy would be very helpful.

If it is not possible to plot it, adding a statement on how long it takes would help.

3. Minor change: Some of the figures can benefit with titles

4. I might need some clarification on the training reward plots. I can see that in the pendulum-swing-up the curve for SINDy-RL starts at a later time-step to account for the data/interactions offline. This does not seem like the case for swimmer and cylinder.

Do the training plots account for the data collected offline? Can that be clarified in the paper? If they do not account for the offline interactions---why so? A clarification on either way would be great (in the reviews and the paper).

If I missed the part in the paper this is written, could it be added to the description of Figure 2?

(Remarks on code availability)

Reviewer #3

(Remarks to the Author)

This paper aims to bring to RL ideas that were developed for ROMs using the SINDy approach introduced by the corresponding authors group. These ideas involve developing surrogates for components of the RL algorithm such as the policy and the reward.

I find that the paper is quite thick in details and the algorithms full of meta-parameters that make a proper evaluation almost impossible. If these details were important I can understand their inclusion. However I fail to see what is the purpose here. This comments applies to comparison of the methods with other approaches such as for example baseline PPO which can be used in multiple of ways. There have been several state of the art RL algorithms used for flow control by the groups of Biferale, Hasegawa, Koumoutsakos and Vinuesa. Comparisons with them on challenging benchmark problems would have been very meaningful. In this context the the most confusing of all comparisons is what the authors keep referring to the "cylinder" benchmark and arguing that SINDy RL is the state of the art in the field. One has to search down at the Appendix to find out that the cylinder is a flow at $Re=100$ which is the simplest flow and it is well known that it can be described by an oscillator, that is with two degrees of freedom. Even so as the authors acknowledge for such a simple flow their methodology is struggling (section 3.4). I believe the authors know well that flow past a cylinder at $Re=100$ is not representative of bluff body aerodynamics. More challenging tests are necessary to showcase their approach.

Suggested revisions:

- Can the authors demonstrate generalisation? One of the drawbacks of SINDy is that is rather well tuned to its training data. In flow control one often aims for the element of "surprise" as the controlled flows are often not in the same phase space as their uncontrolled counterparts. Can the authors demonstrated that the learned policies can be transferred to some unknown regimes and their interpretable results carry over to these unknown regimes as well?
- How about comparisons with existing works in fluid mechanics (see groups above or works from others)? How about comparisons with the works of Fukagata and Hasegawa in cylinder flows (referenced in the paper)? Can they also demonstrate control of challenging flows (like control of 2D cylinder at Re greater than 10,000 or a 3D cylinder above 1000, or a turbulent flow)?

If the authors can address the above two concerns the paper would be worthy of publication. In its present form the article introduces rather ad-hoc surrogates for RL operators. I find that the paper is difficult to read, involving extensive meta-parameter tuning and does not include any convincing results that advance the state of the art of flow control and its interpretation.

I regret that I cannot recommend publication of the paper in its present form.

(Remarks on code availability)

Version 1:

Reviewer comments:

Reviewer #1

(Remarks to the Author)

Now the draft looks better, only one thing which I think it is better to clarify it for the readers is this part:

``In this work, we simply set a fixed number of dynamics updates, but in practice one would either stop training once the performance has plateaued or reached some task-specific criteria determined by the practitioner. We have clarified this in the draft."`

(Remarks on code availability)

Reviewer #2

(Remarks to the Author)

The new draft of the manuscript is great! I really like the inclusion of the fluidic pinball with 70 dimensions! It drives the purpose of the paper very well in a more real-world environment.

I had three minor comments on: including clock time information, longer captions for each figure to make it informative, and titles for missing figures. All of these have been addressed. The longer captions on the figures were very helpful.

All my comments and concerns have been addressed!

(Remarks on code availability)

Reviewer #3

(Remarks to the Author)

I believe that the argument if it works in atoy problem it is useful for solving complex problems is not valid any more in 2025.

(Remarks on code availability)

I thank the authors for considering my comments. However, the core of my arguments has not been answered. The authors make major claims about their method when it is only performing at toy problems.

Today RL has been used to model complex 3D flows (cylinders, channels, boundary layers)

so I fail to understand the merit of a methodology that is only shown to work in simple 2D flows and toy environments.

The authors present no evidence that the methodology is potent enough to solve complex, non-linear problems.

The authors show again flows at Re around 100. The 2D flow past a single cylinder at Re up to 500 have the same dynamics as $Re=100$ - one can verify this by projecting the dynamics in a two-three dimensional latent space. And the "pinball" flow with the arrangement of three cylinders does not add anything in terms of complexity or non-linearity as the dynamics amount to a linear superposition of linear dynamics of a single cylinder. Finally the additional paper that is brought forward by Wolf et al. also shows simple dynamics including for a example a lid driven cavity flow in the laminar regime.

I regret that I maintain my objections and cannot recommend publication of this paper.

Version 2:

Reviewer comments:

Reviewer #2

(Remarks to the Author)

All my concerns have been addressed. I appreciate the paper including another higher dimensional domain in the 3D Airfoil, including its accompanying analysis. I believe this makes the paper stronger.

(Remarks on code availability)

Response to referee comments on “SINDy-RL: Interpretable and Efficient Model-Based Reinforcement Learning”

Dear Referees,

We are grateful for your careful reviews and for the helpful suggestions. These comments have provided us with valuable perspectives and have helped improve our arguments considerably. In particular, your comments motivated us to investigate how our methods perform on a more complicated fluid environment and examine how well trained agents generalize to different, unseen dynamic regimes. We have added a fourth environment, the fluidic pinball, which we believe to be significantly challenging because: (1) it is governed by the Navier-Stokes equations and is computationally intensive to simulate, (2) the state is only partially observable and uses a 70-dimensional observation space, (3) the reward is not analytically computable from the observation, (4) the dynamics are much more complex than the flow past a cylinder, and they undergo a series of bifurcations as the Reynold’s number Re is increased. The fluidic pinball has become a more complex benchmark fluid flow control problem, as it has tunable complexity as a function of Re , exhibiting more rich and chaotic dynamics.

Consistent with our previous results, we are able to learn both a dynamics model and reward to train agents capable of controlling the system and improve the sample efficiency by a factor of about $15\times$. This greatly reduces the amount of time we need to train the agent to just ~ 6.5 hours (and ~ 7 hours of initial off-line data collection) compared to nearly 6 days of training an agent in the full-order environment—providing evidence that our method could be used to rapidly accelerate a practitioner’s ability to build sophisticated controllers in expensive environments.

The large observation space provides a common challenge for using DRL for real-life applications. A recent preprint, “Interpretable and Efficient Data-driven Discovery and Control of Distributed Systems” by Wolf and collaborators was made available after our initial submission to Nature Communications. The authors apply our SINDy-RL method to PDE control using large state and action spaces by using an autoencoder simultaneously learn a nonlinear projection into a low-dimensional space; providing further evidence that our method can be extended to high-dimension optimization problems. To complement this recent work, we use a fixed linear projection onto the 10-dominant SVD modes, providing a simple alternative that does not require changing the coordinate representation throughout training.

Finally, the bifurcation allows us to examine the generalizability of our method. By training as $Re = 100$, the agent only experiences dynamics with quasi-periodic vortex shedding, for Re greater than ≈ 115 , the dynamics become chaotic. We evaluate our agent at $Re = 150, 250, 350$, and demonstrate that the projection of a neural network policy onto the dictionary policy (with nearly two orders of magnitude fewer parameters) improves the ability to generalize significantly. This again provides further evidence for our ability to extract compact, symbolic representations from black-box functions, such as neural networks,

that can operate in power- and memory-constrained environments.

By incorporating these new contributions and your other feedback, we believe that this latest draft provides a much stronger argument for using our method to provide lightweight, symbolic models with reinforcement learning for applications in engineering and the applied sciences. Below, you will find a detailed summary of how each review was addressed. Thank you again for considering this work, and we look forward to hearing feedback about our revision.

Best regards,

Nicholas Zolman, Urban Fasel, J. Nathan Kutz, and Steven L. Brunton

Referee 1

General Author Response: Thank you for your careful reading of this paper. We believe your feedback has helped clarify different technical points in our draft and overall strengthen our arguments. In terms of major changes, we have taken content from the supplementary file back into the main text to improve the readability of the methods and provide context for the environments. In particular, we have included the SINDy Dyna-style MBRL algorithm into Section 2 and included information about environment tasks into the Figure 2 where the results are summarized.

The current draft needs improvement. It is difficult to always switch to main draft and supplementary file!

Author Response: We appreciate this feedback, and we have made a significant revision to clarify and streamline the presentation.

- What do you mean by "interpretable control policy orders of magnitude"?
 - **Author Response:** In this context, we mean that the sparse dictionary policy is interpretable and has orders of magnitude fewer parameters than a neural network counterpart.
- I think this sentence should be re-written “In this work, we develop interpretable and generalizable reinforcement learning methods that are also more sample efficient via sparse dictionary learning.”

the reason is that “interpretability” means being more specific while “generalization” contradicts it!

 - **Author Response:** We apologize for the confusing wording, and we have clarified this in the text. In this context, “interpretability” and “generalization” are harmonious concepts because they both result from using models with fewer parameters. The sparsity highlights only a few important terms in the models, allowing one to interpret their meaning. Likewise, a model with fewer parameters is provably less prone to overfitting, fostering generalization to unseen data.
- In the Background, this sentence “at each state $u_n\pi(x_n)$ and executing it in the environment, producing a new state x_{n+1} ”, should be edited.
 - **Author Response:** Thank you for the comment, we have made edits to clarify the setup in the text.
- In contributions, what does “Improve sample efficiency by orders of magnitude for training,” mean?
 - **Author Response:** We have clarified this in the main text. “Sample efficiency” refers to the number of collected data samples from the full-order environment.

Our results demonstrate that we can reduce the number of interactions in the full-order environment and achieve comparable performance using $20 - 100\times$ fewer samples.

- Modify this sentence “By leveraging sparse dictionary learning in tandem with deep reinforcement learning,” please do not use “tandem“ here, please consider to use usual words.

- **Author Response:** Thank you for the suggestion. We have replaced this word in the main text.

- Well, it is not always true to say “SINDy” is quite efficient and provide the interpretable models! specially when the size of “library” increases!

One may use another system identification method and fit a model to the given dataset without constructing a big matrix named “library” and use it in practice, like Adaptive control methods. Instead you could say “We tested a new approach, and based on our results performs better than Baseline PPO, etc.”

- **Author Response:** Thank you for the comment; we certainly appreciate the perspective that SINDy has its limitations and we have taken extra steps to make this clear in the main text in order to not oversell our approach. We do not wish to argue that SINDy is the only system identification algorithm that should be used for model-based reinforcement learning. We instead argue that SINDy can often be a convenient method for doing this system identification and comes with a number of benefits, such as efficiency, interpretability, and uncertainty quantification. We also wish to acknowledge the drawbacks of SINDy—as the size of the library grows, the optimization problem becomes significantly harder, and we begin to lose both interpretability and efficiency without other methods to prune the library. In our approach, we took extra steps to mitigate the cost of the large library size by using Ensemble-SINDy with library dropout, which has previously been shown to significantly improve the sparsity for large libraries when taking the median coefficients. For large input spaces that lead to intractably large libraries, we also demonstrate how dimensionality reduction can be used to mitigate this challenge. Finally, we advocate for choosing the simplest libraries to start and increase complexity as needed when the amount of data available makes it feasible to fit.

- In Background section, “a more detailed discussion can **be** found in Supplementary,”

- **Author Response:** Thank you for catching this typo, we have addressed it in the text.

- “The agent interacts **with** the environment,”

- **Author Response:** Thank you for catching this typo, we have addressed it in the text.

- r_n (reward, immediate cost, or regret) is not a signal in RL! it is a function! $X \times U \rightarrow R$
 - **Author Response:** We apologize for the confusing wording, and have addressed this in the text. We did not mean to indicate that the reward is a continuous signal, we simply meant to informally describe the reward as the “learning signal” for the agent.
- Check the related literature in RL/ADP to see how they defined value function/ cost function, etc.
 - **Author Response:** Thank you for the suggestion. We use a definition of the value function consistent with the classic reinforcement learning text by Sutton and Barto [1].
 - [1] Sutton, Richard S., and Andrew G. Barto. Reinforcement learning: An introduction. MIT press, 2018.
- How do you define the satisfactory performance in your algorithm? as you said “We repeat this process and update the dynamics model until the agent has reached satisfactory performance”!
 - **Author Response:** Thank you for the comment. In this work, we simply set a fixed number of dynamics updates, but in practice one would either stop training once the performance has plateaued or reached some task-specific criteria determined by the practitioner. We have clarified this in the draft.
- If you used “D” for “Deep”, then stick to it in the draft — “see deep neural networks” in RL background
 - **Author Response:** Thank you for the suggestion, we have taken steps to address this in the text.
- In section 2, SINDy-RL, “Approximating Dynamics,” you say Dyna-style MBRL algorithm! but I could not find it! in the Supplementary file also is not named! you can even use “itemize” in latex to make it better here!
 - **Author Response:** Thank you for the suggestion. We have moved the algorithm from the supplementary file into the main text to help improve the readability of the section.
- $y = \dot{x}$, introduce \dot{x} or just say $f(x)$ as you did earlier!
 - **Author Response:** Thank you for the suggestion, we have taken steps to address instances of \dot{x} in the text.
- In equation (1), what is F! please consider that your reader may not know what is “norm” at all!

- **Author Response:** Thank you for the suggestion, we have clarified it to be the Frobenius norm in the text.
- Well, E-SINDy is not necessary robust in the low-data limit!
 - **Author Response:** Thank you for the comment. In the text, we specify that E-SINDy is *more* robust than SINDy in the low-data limit. This is generally true of ensembling methods compared to their single-model counterparts.
- Please elaborate on this “Because most dynamic systems (especially with polynomial dynamics) diverge, we bound the state space and reset the surrogate environments during training if a trajectory exits the bounding box.”
 - **Author Response:** Thank you for the suggestion, we have expanded this in the main text. For a polynomial dynamical system, $dx/dt = p(x)$, where $p(x)$ is a polynomial of x , the existence of a solution is only guaranteed for some small amount of time, $t_{\max} = \epsilon$ before the solution diverges. Quadratic dynamics are particularly susceptible to divergence because the x^2 term will dominate if $|x| > 1$, ensuring that the dynamics dx/dt will diverge to $\pm\infty$. To handle the problem of diverging dynamics, we place an artificial bound on the state space and end the simulation if a trajectory escapes these bounds. In the discussion section, we discuss how one could alternatively enforce the stability of the dynamics within a trapping region using semi-definite programming.
- Please elaborate on this sentence “Instead of only sampling points from the surrogate trajectories, we can sample plausible points in a neighborhood surrounding them to create a more accurate approximation while still avoiding regions of the space that can no longer be trusted.”
 - **Author Response:** This is a good point. We have expanded this in the main text. When performing imitation learning, there is a question of what trajectories one should use to train the policy. While one could generate a significant amount of data by running the expert policy (i.e. a fully trained DNN) in the full-order environment, it can be prohibitively expensive. To cheaply generate plausible data, we sample trajectories generated by the expert policy using the learned dynamics model. Because we only learn a low-fidelity SINDy model, the dynamics are generally more sensitive to control inputs than the true model and trajectories with different initial conditions can quickly converge to the same point—once again, this risks over-fitting to a particular region of phase space. To combat this, we sample plausible points in a neighborhood surrounding these surrogate trajectories by adding noise (which can also be viewed as random disturbances), resulting in a more robust controller. In the supplementary materials, we evaluated a number of different ways to sample these trajectories.
- You say “x-direction as possible in a fixed time,”! x was your state!

- **Author Response:** Thank you for the comment, we have clarified it to be the horizontal direction to alleviate confusion about notation.
- The word of “surrogate“ it is fuzzy and to tell the truth and hard to digest! you may say one line in introduction about it!
 - **Author Response:** We have clarified the use of the word in the introduction.
- In the Benchmark section, you introduce several case studies! the reader needs to see in brief their state space x , control inputs u , etc. I saw in the supplementary file “Figure-2“ gives a brief summary of the dynamical systems! so put such figure in the draft! the reason is that reader may not know the dynamic system and simply can have a look at the equations and get the problem!
 - **Author Response:** We appreciate this suggestion. We have included information from the supplementary figure you reference into Figure 2 in the main text to make this more clear for readers that are not familiar with the different dynamical systems.
- One important question that raise is “all the case studies are deterministic, i.e. give a state x_k , and control u_k then there is a unique x_{k+1} ! so why do you say “stochastic policy?“ and if so bring an example such as Markov Decision Processes!
 - **Author Response:** Thank you for the question. You are correct; in this manuscript, we only provide results for deterministic environments. However, the MBRL framework we introduce is meant to be agnostic of the policy optimization algorithm, \mathcal{A} , which may update the parameters for a stochastic policy. In addition, while deterministic policies can be used for a deterministic environment, stochastic policies are also useful in exploring the control space during training—the so called “exploration-exploitation” trade-off. We have taken steps to clarify this in the text.
- Now lets focus on one case study, e.g “Cartpol swing up”, its state space is defined as $(x, \cos(\theta), \sin(\theta), \dot{x}, \dot{\theta})$, of course we can give values to these and create a large state space! but as we all know in practice engineers do not do such naive strategy! this happens in almost in entire industry! such as power plants, motors, etc.

Indeed we say let’s consider a “practical operating point, e.g. $\theta = \frac{\pi}{2}$ then linearize this non-linear model and design a controller for it! this gives a simple, interpret-able, feasible and applicable! or we apply directly nonlinear control methods with guaranteed stability!

Now why should I choose an arbitrary policy π_0 and explore “unsafe” state space and meet non-real and non-sense environment?

By the way, why do not you employ “projection based RL approach“ such as Temporal Difference, LSTD, etc.? One can take $(x, \cos(\theta), \sin(\theta), \dot{x}, \dot{\theta})$ as feature vector and

project the large state space into the selected features and use Monte Carlo simulations and policy iteration or policy evaluation to compute a vector which can be used as the optimal value/cost function!

why do not you compare the results with conventional methods such as “linear” or “non-linear” or “adaptive” control methods?

- **Author Response:** Thank you for the discussion; we certainly agree with you—if it is possible to use a simpler method with robustness guarantees, then one should consider using that method before trying a more complicated approach with fewer guarantees. In the context of this work, we are assuming that we do not have access to a model of the state dynamics to start with, so it is not possible to linearize around a fixed point because we do not have access to the nonlinear dynamics at that point. Furthermore, in the fully-nonlinear regime away from the desired state, linearized methods are known to fail—e.g., this has been thoroughly studied in the case of cylinder in a fluid flow for $Re > 80$, which applies to our example at $Re = 100$.

As stated above, the approach we introduce is meant to be agnostic of the policy update algorithm, \mathcal{A} . As you suggest, this could be achieved through temporal difference, LSTD, and policy iteration using the surrogate model of the dynamics to estimate the value function. For each choice of algorithm \mathcal{A} , we expect that learning a policy using trajectories sampled from surrogate SINDy dynamics should be achievable using fewer trajectories from the full-order environment than learning a policy using only trajectories from the full-order environment alone. This was also shown to be the case previously, where SINDy-MPC was used to produce Monte-Carlo simulations for determining the optimal control input. In this work, we chose to limit our scope where \mathcal{A} was the PPO algorithm and we compared different approaches to identify the dynamics models, such as linear and DNN dynamics models.

- [1] Illingworth, Simon J., Hiroshi Naito, and Koji Fukagata. "Active control of vortex shedding: an explanation of the gain window." *Physical Review E* 90.4 (2014): 043014.

- Please include the processor that you used to simulate different case studies.
 - **Author Response:** We have included the compute specifications that we used for both the rollout of the environments in training in the text. All experiments were performed using a single-node, Linux engineering workstation consisting of a total of 40 CPU cores (Intel[®] Xeon[®] Gold 6230). No GPUs were used during training or simulating the environments.
- Since you use DNN, it is recommended to cite recent works, such as : -“DeepMoD: Deep learning for model discovery in noisy data,” *Journal of Computational Physics*.

- “A Robust SINDy Approach by Combining Neural Networks and an Integral Form,”
<https://arxiv.org/abs/2309.07193>.

- **Author Response:** Thank you for the suggestion. While we do not use DNNs for the dynamics or reward model discovery, we agree that it is important to include references for alternative approaches for model discovery. We have included these references into both the main text and the supplementary file containing the extended background.

Referee 2

General Author Response: Thank you for the thoughtful feedback and careful reading of our paper. Your suggestions encouraged us to investigate our approach on a higher dimensional problem, and we now have extensive results applying our method to the fluidic pinball environment consisting of three independently rotating cylinders in a fluid flow. We believe this to be a much more challenging optimal control problem due to being partially observable, the size of its observation space, and physical separation between the reward and sensing information. Furthermore, the presence of a bifurcation for increasing Reynold’s number provides a definitive boundary for evaluating how the agent extrapolates to unseen dynamics. We believe that our success on this challenging problem and our ability to generalize beyond the dynamics seen during training significantly strengthens our presentation of these methods. In addition, we have also investigated the suggested clock times of our two flow control problems. For both examples, the iterative training process for SINDy-RL is dominated by experience collection by an order of magnitude; thus the reported time efficiencies are comparable to the sample efficiencies. We provide a full discussion to your suggestions below.

This paper combines advances in physics and machine learning by introducing SINDy-RL—a method that combines sparse learning of dynamics and reinforcement learning. SINDy-RL is a Dyna-style algorithm for Model-Based Reinforcement Learning which fits the dynamics model of the world, the reward function, and the policies.

This work is novel as compared to past work as it also learns dynamics models for the reward function and policies. Learning the reward function is useful in complex systems, and learning a representation of the policies can help in time-sensitive applications where large neural networks are slow (although learning a linearly-combination of non-linear terms to express a policy can cause in loss of accuracy).

This paper has merit since the authors compare their learned policies against the ground truth dynamics functions—providing evidence that on the working of the method.

I believe the paper can be improved with the following tweaks:

1. Currently, the environments being tested against are low-dimensional. Particularly, pendulum-swing-up and swimmer are lower dimensional environments. These environments have their merits since their lower-order allows for easier explainability. However, results on a higher-order environment will solidify my trust in the methods working. Examples could include Hopper, Halfcheetah, or more. I recommend adding these results in.

Author Response: Thank you for the suggestion; we certainly see the merit of demonstrating this approach on higher dimensional environments. We have taken steps to address this by evaluating our method on the more challenging fluidic pinball environment. This objective seeks to minimize the net drag exerted on a system of three independently actuated cylinders in a fluid flow from 35 x - and y - velocities in the wake. This environment is particularly challenging because:

1. the dynamics are governed by the Navier-Stokes equation and is computationally intensive to simulate

2. the state is only partially observable and uses a 70-dimensional observation space
3. the reward is not analytically computable from the observation, and information must flow downstream to the sensors
4. the dynamics undergo a bifurcation as the Reynold’s number, Re , is increased.

The first challenge highlights the usefulness of our approach at saving time for practitioners. By offloading the experience collection to the surrogate SINDy model, we can greatly reduce the physical amount of time training the policy. As indicated in our new draft, the baseline DRL method took 6 days of training time, whereas the SINDy-RL approach only took 6.5 hours. Even accounting for the additional 7 hours of initial data collection to initialize a surrogate environment, our approach uncovered a sophisticated policy much faster; thus making it practical to rapidly discover and iterate on control policies for complicated, computationally expensive environments.

The second challenge, high-dimensional observation spaces, highlights a common issue that practitioners encounter when seeking to utilize DRL. The high-dimensionality of the problem makes it challenging for any optimization problem to learn a model—whether it is a surrogate dynamics, reward function, policy, or even the value/ Q – functions used in policy optimization; it is therefore critical to consider dimensionality reduction approaches. In our approach, we provide a simple, linear projection onto the leading 10-dimensional SVD modes and use this to train both the SINDy-RL and baseline agents, providing a fair comparison between the two methods. Even with this projection, this is the largest observation space that we have evaluated on yet, and the size of the optimization problem is directly comparable to that of the Hopper environment you suggested (with an 11-dimensional observation space and a 3-dimensional action space). We believe our success in training agents using this reduced set of coordinates provides a convenient and practical path for scientists and engineers looking to apply our methods. We would also like to recognize a recent work [1] that combined our SINDy-RL approach with SINDy autoencoders for discretized PDE control after our initial submission to Nature Communications. Motivated by the discussion in our arXiv preprint, the authors used autoencoders to discover nonlinear coordinate projections for both the observation and action spaces into a low-dimensional latent space and successfully use our SINDy-RL approach to learn efficient controllers. This provides another, complementary path for learning a low-dimensional coordinate system to model the dynamics and provides further evidence for the efficacy of our approach to high-dimensional nonlinear systems.

The third challenge comes from the fact that our sensing information does not directly capture information needed to model the reward. Whereas the sensors (and therefore their SVD projection) provide information in the wake downstream from the rotating cylinders, the net drag force is localized to the flow on the surface of the cylinders. Despite this, our dictionary learning approach finds a sufficiently descriptive representation of the reward function by coupling information from the control inputs and information from the wake. This result highlights the practicality of our method for measurement reconstruction, which is common throughout engineering applications.

The fourth challenge highlights the importance of generalizing to new, unseen dynamics. In many physical systems, the physical parameters of the system are only known with some degree of certainty; if an agent is trained in a simulation environment and overfits to the observed dynamics from a particular set of parameters, then it may not transition to the real environment it is being designed for. Furthermore, as we have discussed, it is expensive to train DRL agents; any ability to transfer knowledge between environments is hugely important in reducing the amount of computation. Our successful demonstration of generalizing to a fully chaotic regime further strengthens our arguments for the usefulness of dictionary policies.

- [1] Wolf, Florian, et al. "Interpretable and Efficient Data-driven Discovery and Control of Distributed Systems." arXiv preprint arXiv:2411.04098 (2024).

2. Refitting to SINDy with large amounts of data can be expensive. While the model itself may not be computationally intensive as compared to a neural network, the training process could take longer.

A plot that compares the reward vs the clock time would be beneficial. For instance, it can take vanilla PPO 3 hours to reach an optimal reward on a task, and it can take SINDy-RL 6 hours to do the same—while SINDy-RL took less interactions with the environment, vanilla PPO took lesser clock time.

In environments where it is very difficult to interact with the environment, such as robotics, I would choose SINDy-RL. In cases where interactions with the environment are not costly, I would choose the neural network and then prune it.

Having a clock time comparison makes the argument stronger. Knowing more about the time it takes to fit a SINDy model when adding additional data on-policy would be very helpful.

If it is not possible to plot it, adding a statement on how long it takes would help.

Author Response: Thank you for the suggestion, we can certainly appreciate the concern about computational trade-off. While SINDy is a form of symbolic regression, the linear structure lets us perform regularized linear least squares regression, which has extremely optimized implementations. In particular, we use sequentially thresholded ridge regression (STRidge) which solves smaller L^2 -regularized least squares problems with each iteration. In general, though, for large libraries (e.g. polynomials on a large-dimensional observation space), the size of the optimization grows significantly and so do the number of resources (though techniques in randomized linear algebra can help alleviate this). For the size of our problems considered, we did not encounter this. For our largest environment, it took on the order of seconds to fit the ensemble of 20 dynamics models *serially*. This was performed using `scikit-learn`'s linear regression packages; for real-time operations this could be further optimized by compiling the functionality in a lower-level language and through parallel implementations of the ensemble fits.

It should be noted that we also create a data buffer which a maximum size—pruning old experience as new experience is collected. The motivation for this was simply to balance the

initial offline collection and newly collected data in order to not bias the learned models too heavily towards the optimally controlled trajectories—e.g., if the system is quickly driven towards the desired state, then the dynamics should remain zero for the majority of the trajectory. For resource constrained and/or real-time operations, the size of the data-buffer could be chosen based off the computational budget available for the linear regression.

In Supplementary Table 4, we have added a table indicating the approximate clock time for various components of training the fluids environments with SINDy-RL. We find that experience collection in the full-order environment dominates significantly, followed by the amount of time used for updating the policy between full-order collections. It is suspected that the amount of time used to update the policy could be significantly decreased since we did not tune this parameter; for real-time operations the amount of time spent training the policy should again be chosen based on the requirements of the particular application and constraints.

Ultimately for the Pinball example, we spend about 7 hours collecting initial data offline, then about 6.5 hours of iteratively training the policy and collecting experience from the full-order environment to reach the same performance that the baseline approach needed 6 days for.

3. Minor change: Some of the figures can benefit with titles

Author Response: Thank you for the suggestion, we’ve taken steps to make the figures clearer by including additional titles.

4. I might need some clarification on the training reward plots. I can see that in the pendulum-swing-up the curve for SINDy-RL starts at a later time-step to account for the data/interactions offline. This does not seem like the case for swimmer and cylinder.

Do the training plots account for the data collected offline? Can that be clarified in the paper? If they do not account for the offline interactions—why so? A clarification on either way would be great (in the reviews and the paper).

If I missed the part in the paper this is written, could it be added to the description of Figure 2?

Author Response: Thank you for the question, we have clarified this in the caption of Figure 2. Supplementary Table 2 lists the original off-policy collection before training the environments. For **swing-up**, we use 8,000 training samples to fit the first dynamics model, this is two times the number of samples used for the policy update (4,000) for the vanilla PPO; this explains why the red curve (vanilla PPO) starts before the blue curve (SINDy-RL) in Figure 2 and Figure 3. For the Swimmer, there was 12,000 experience collected to fit a dynamics model (per member of the population), which is cut off from the scale in Figure 2 in the main text (chosen for clarity in the size of the plots), but can be seen in the full plot in Supplementary §4. The Cylinder used 3,000 experience up front, which is less than the amount of one policy update used for the vanilla PPO implementation (4,000). This offset can be seen in Figure 2 in the main text with the blue curves (SINDy-RL) starting on the x-axis slightly before the red curves (vanilla PPO).

Referee 3

General Author Response: Thank you for your careful reading and insightful feedback. In particular, your comments have motivated us to (1) simplify the presentation of our methods and results, (2) incorporate an additional, challenging fluids environment, and (3) demonstrate the ability to generalize to dynamics regimes beyond the training data.

This paper aims to bring to RL ideas that were developed for ROMs using the SINDy approach introduced by the corresponding authors group. These ideas involve developing surrogates for components of the RL algorithm such as the policy and the reward.

I find that the paper is quite thick in details and the algorithms full of meta-parameters that make a proper evaluation almost impossible. If these details were important I can understand their inclusion. However I fail to see what is the purpose here. This comments applies to comparison of the methods with other approaches such as for example baseline PPO which can be used in multiple of ways.

Author Response: Thank you for the helpful feedback. We have taken steps to simplify both the presentation of our methods and results. For the hyperparameters, we would like to distinguish between three different classes: (1) the “new” hyperparameters we introduce for Dyna-style MBRL in Algorithm 1, (2) existing hyperparameters for the model-free DRL algorithms, such as PPO, and (3) hyperparameters for the surrogate models, such as SINDy.

For the new hyperparameters at the core of our Dyna-style MBRL, we introduce three very basic parameters ubiquitous across all Dyna-style methods: how much data you collect to initialize a surrogate model (N_{off}), how long you train the policy before evaluating it (n_{batch}), and how long you collect data during evaluation ($N_{collect}$). For real applications, the choice of these parameters are often dictated by *a priori* constraints—such as time or resource budgets for an application, the amount of available data available from a previous experiment, or a characteristic time for performance evaluation. In the supplementary file, we perform a deep investigation into the dependence of the algorithm’s performance on n_{batch} and $N_{collect}$ since practitioners generally have more control over these parameters. We find that there is a wide choice of different parameter combinations such that SINDy-RL provides substantial improvements to the sample efficiency and that generally it is the ratio of these two hyperparameters that matter—indicating that there may simply be a single practical choice that needs to be made.

For the DRL hyperparameters, we recognize the challenge of performing a fair and tractable comparison when a DRL algorithm may have many; any sort of hyperparameter sweep become computationally infeasible rather quickly. To account for this, we choose to fix the set of PPO hyperparameters across all experiments—i.e. across all four tested environments and all model-free and model-based benchmarks that we used. As described in our supplementary file, the hyperparameters were chosen to be most similar to the CleanRL’s exhaustive benchmarks. From the perspective of a practitioner pursuing a novel application, using this set of hyperparameters as a starting point is justifiable when one cannot afford to perform a hyperparameter sweep. Furthermore, to fairly compare across

different instantiations of neural network policies, we report the statistical performance across 20 independently seeded experiments.

Finally, when we compare the comparison of different Dyna-style algorithms in our benchmarking section, the primary purpose is to demonstrate why one would use SINDy as a dynamics model instead of something simpler (such as linear dynamics) or more complicated (such a neural network). To create as fair of a comparison between the three types of dynamics models, we fix the Dyna-style algorithm and simply swap out dynamics models. In the case of the SINDy and linear models, there are only two choices for hyperparameters: the L^2 regularization coefficient and the sparsity threshold. For small dictionary models (like the ones we consider), the methods are insensitive to the regularization coefficient since it mostly plays a role when the models are ill-conditioned, such as being under-determined due to lack of data. The sparsity threshold certainly plays an important component, however it can easily be chosen via cross-validation using a line-search and pareto-front analysis. The neural network has many more hyperparameters, including its choice of architecture and optimizer. We tried to make as fair of a comparison as possible using networks sizes and architectures seen in the literature for similarly-sized problems. However, the sheer number of choices to be made again demonstrates the desirability to use SINDy and its two hyperparameters.

We would also like to note that while hyperparameter tuning is made generally infeasible for resource-intensive environments (such as CFD or laboratory settings), the speed efficiency gained by using SINDy-RL may make this more tractable. In the text, we demonstrated the ability to perform hyperparameter tuning with population-based training to quickly identify tailored sets of hyperparameters. We have restructured the benchmarking section to highlight this utility.

There have been several state of the art RL algorithms used for flow control by the groups of Biferale, Hasegawa, Koumoutsakos and Vinuesa. Comparisons with them on challenging benchmark problems would have been very meaningful. In this context the the most confusing of all comparisons is what the authors keep referring to the "cylinder" benchmark and arguing that SINDy RL is the state of the art in the field. One has to search down at the Appendix to find out that the cylinder is a flow at $Re=100$ which is the simplest flow and it is well known that it can be described by an oscillator, that is with two degrees of freedom. Even so as the authors acknowledge for such a simple flow their methodology is struggling (section 3.4). I believe the authors know well that flow past a cylinder at $Re=100$ is not representative of bluff body aerodynamics. More challenging tests are necessary to showcase their approach.

Author Response: Thank you for the comments; we appreciate your feedback. We would like to clarify that we are not arguing that SINDy-RL is achieving state-of-the-art *performance* among all possible environments—we are instead providing demonstrations that the SINDy-RL can help accelerate the training process to achieve *comparable* performance to state-of-the-art *algorithms*. Indeed, one would not expect that approximations of the dynamics should lead to better performance on a particular environment, except in cases where one is interested in meta-learning and generalizing to new physical parameters of a dynamic system; then perhaps the policy has been introduced to a wide-variety of similar

(but different) dynamic systems and is more robust to all.

We certainly appreciate the perspective that the Cylinder at $Re = 100$ is too simple of an environment. We chose it because it had been well-studied and HydroGym developed an open-source implementation to be used with reinforcement learning. We restrict our attention to $Re = 100$ due to constraints regarding the mesh and actuation. We reiterate that we do not claim that we have achieved state-of-the-art performance on this task, simply that we use fewer interactions in the full-order environment to perform comparably with PPO, and the range of results is comparable with what is found in the literature; we have taken steps to make this more clear in the text and not oversell our method. We would also like to clarify that in Section 3.4, we were highlighting that our methodology for fitting a sparse representation of the *neural network policy* struggled, not the Dyna-style methodology in general. We have taken steps to clarify this in the text and have included a small discussion that this is due to the policy being well-represented by a discontinuous “bang-bang” control law, which is not well approximated by polynomials.

Importantly, we have added a new, more complex fluid benchmark problem, the fluidic pinball environment from HydroGym. This has become an accepted benchmark problem that is more complex than the flow past a cylinder, but is still accessible with reasonable computational hardware. We believe that the open-source nature of the environment will allow others to replicate and build upon our results. The environment itself is significantly more challenging, even at $Re = 100$, due to the increased observation size and actuation space—increasing the need for exploration when sampling the policy. Further, due to a bifurcation at $Re \approx 115$ when the dynamics become chaotic, it offers a natural opportunity to explore the generalization of our technique. We have demonstrated strong performance on this benchmark, as well as promising generalization, as discussed below.

Suggested revisions: - Can the authors demonstrate generalisation? One of the drawbacks of SINDy is that it is rather well tuned to its training data. In flow control one often aims for the element of “surprise” as the controlled flows are often not in the same phase space as their uncontrolled counterparts. Can the authors demonstrate that the learned policies can be transferred to some unknown regimes and their interpretable results carry over to these unknown regimes as well?

Author Response: Thank you for the recommendation; we agree that generalizing to unseen states or unknown dynamics is something of extreme practical importance, and an area where DRL has struggled significantly. To highlight this, we examine how the pinball agent extrapolates to higher Re . Because at $Re \approx 115$, the dynamics experience a bifurcation, transitioning from asymmetric, quasi-periodic vortex shedding to chaos, there is a distinct difference in the observed dynamics and states. We evaluate the pinball agents at $Re = 150, 250,$ and 350 for a trajectory of 100s after only training on data from $Re = 100$ on trajectories of 20s. We find that when evaluating the neural network policy obtained from the SINDy-RL Dyna-style method, the agent is capable of greatly reducing the net drag and nearly stabilizing the wake. Importantly, we demonstrate that the sparse dictionary policy (obtained from its neural network counterpart) obtains significantly better generalization capabilities, completely suppressing vortices in the wake. The ability to zero-shot generalize

to the fully-chaotic regime at $Re = 350$ highlights the important role that parsimony through a learned dictionary policy can play in the generalizability of control strategies.

- How about comparisons with existing works in fluid mechanics (see groups above or works from others)? How about comparisons with the works of Fukagata and Hasegawa in cylinder flows (referenced in the paper)? Can they also demonstrate control of challenging flows (like control of 2D cylinder at Re greater than 10,000 or a 3D cylinder above 1000, or a turbulent flow)?

Author Response: Thank you again for recommending we try this on a more difficult example. As previously mentioned, we sought to apply our methods to the fluidic pinball environment available in HydroGym at moderate $Re = 100$ and evaluated in the fully chaotic regime up to $Re = 350$. We recognize that this example may not be as challenging as the $Re = 1,000$ 3D or $Re = 10,000$ 2D flows. While we are engaging in collaborations for more focused and challenging applications, we scope this work primarily as a methods paper; as such we believe as a first publication it is most important to choose examples that are easily reproducible by the average practitioner without having to develop their own trusted simulation and control framework. The suggested environments at high Re require significant computational (or laboratory) resources that are unavailable to most. Furthermore, we sought an open-source implementation that has already been validated for use in control settings—particularly DRL—where the policy may break the solver without significant care or mesh-refinement. We believe the Pinball environment to be a good middle-ground; it (1) is significantly more challenging than our previous examples, (2) highlights a number practical challenges one might face and how our method could be used in spite of them, and (3) is free, open-source, and runnable on a single machine without need for HPC resources.

If the authors can address the above two concerns the paper would be worthy of publication. In its present form the article introduces rather ad-hoc surrogates for RL operators. I find that the paper is difficult to read, involving extensive meta-parameter tuning and does not include any convincing results that advance the state of the art of flow control and its interpretation.

I regret that I cannot recommend publication of the paper in its present form.

Author Response: We appreciate your constructive and direct feedback, and we believe your suggestions have significantly improved the revised paper. Our goal is not to demonstrate scaling to the largest fluid flow control systems, where only a few groups in the world have the computational resources to implement RL. Instead, we present a new approach for surrogate modeling that should reduce the number of expensive calls to the environment and potentially improve generalization. We are currently working with collaborators to investigate the scaling of this approach to much larger scale problems, although we believe this is beyond the scope of this work. That said, in the revision we have tackled a much more complex new fluids example and demonstrated good generalization, which we believe have significantly strengthened the paper. Thank you again for your helpful feedback.

Response to referee comments on “SINDy-RL: Interpretable and Efficient Model-Based Reinforcement Learning”

Referee 1

Now the draft looks better, only one thing which I think it is better to clarify it for the readers is this part:

“In this work, we simply set a fixed number of dynamics updates, but in practice one would either stop training once the performance has plateaued or reached some task-specific criteria determined by the practitioner. We have clarified this in the draft.”

Author Response: Thank you again for your careful reading of this paper, and we are pleased to hear that you believe our last update was a significant improvement. We appreciate your latest comment, and we have updated the latest draft to have a clearer description of the method.

The most significant change to the latest draft comprises a new environment to test our method (gust mitigation for a 3D airfoil). We include these results while making minimal changes to the main text to maintain the clarity from our previous draft.

Referee 2

The new draft of the manuscript is great! I really like the inclusion of the fluidic pinball with 70 dimensions! It drives the purpose of the paper very well in a more real-world environment.

I had three minor comments on: including clock time information, longer captions for each figure to make it informative, and titles for missing figures. All of these have been addressed. The longer captions on the figures were very helpful.

All my comments and concerns have been addressed!

Author Response: Thank you again for your excellent feedback during our first revision. We believe that it made our previous draft significantly stronger, and we are pleased to receive your latest enthusiastic response! In our most recent draft, we continue to demonstrate the effectiveness of our method on a new aerodynamics problem (gust mitigation of a 3D airfoil in an unsteady flow). Whereas the previous result from the Pinball was on a 70-dimensional observation space, the new environment has a 318-dimensional space that we project down into a lower dimensional space before training our agent; further demonstrating the scalability of our method. As a 3D fluids environment, it was also significantly more expensive to run. In addition to including the new clock times in our SI table, we also discuss the resources needed to run this example. To efficiently run the environment, we needed to distribute the simulation across four NVIDIA A100 GPUs on an HPC node, requiring at least 75GB of VRAM, whereas simulating the surrogate model is nearly negligible and only requires access to CPUs. This setup provides conditions closer to what modern practitioners face when using DRL for aerodynamics and significantly strengthens our claims.

Referee 3

I thank the authors for considering my comments. However, the core of my arguments has not been answered. The authors make major claims about their method when it is only performing at toy problems. Today RL has been used to model complex 3D flows (cylinders, channels, boundary layers) so I fail to understand the merit of a methodology that is only shown to work in simple 2D flows and toy environments.

The authors present no evidence that the methodology is potent enough to solve complex, non-linear problems. The authors show again flows at Re around 100. The 2D flow past a single cylinder at Re up to 500 have the same dynamics as $Re=100$ - one can verify this by projecting the dynamics in a two-three dimensional latent space. And the "pinball" flow with the arrangement of three cylinders does not add anything in terms of complexity or non-linearity as the dynamics amount to a linear superposition of linear dynamics of a single cylinder. Finally the additional paper that is brought forward by Wolf et al. also shows simple dynamics including for a example a lid driven cavity flow in the laminar regime.

I regret that I maintain my objections and cannot recommend publication of this paper.

Author Response: Thank you again for the careful reading of our paper. We sincerely appreciate your constructive feedback, and we recognize the need for demonstrating modern methods on challenging engineering problems. We have taken your feedback seriously, and **we have included an example of controlling the fully unsteady flow around a 3D NACA 0012 airfoil** at $Re = 1000$ for the purposes of gust mitigation.

In response to your feedback from our first draft, we chose the airfoil geometry to be the standard NACA 0012 because of its prevalence in the fluids community for a wide-range of engineering applications. The geometry also induces symmetry-breaking in the flow, and we believe it to be a more realistic representative of bluff-body dynamics compared to a cylinder.

The flow is simulated at $M = 0.2$ with a 20° angle-of-attack, well-within the post-stall regime (typically $14 - 16^\circ$ for NACA 0012), with gust factor $G = 2.0$. Under these conditions, the flow exhibits complex, unsteady dynamics—exhibiting 3D instabilities that are not possible in 2D flows. In addition to creating an efficient implementation of the environment, we have thoroughly validated the CFD solver (and have included a comparison to the literature in the supporting materials) and have verified the robustness of the numerics to stochastic control from an agent actuating the jets.

The scale of the environment is also much closer to what engineers might practically face when seeking to use DRL for flow control with CFD, requiring 72 million cells with the lattice Boltzmann method. To efficiently run the environment, we distributed the simulation across four NVIDIA A100 GPUs on an HPC node, requiring at least 75GB of VRAM, whereas simulating the surrogate model is nearly negligible and only requires access to CPUs. Consistent with our previous results, our method utilized far fewer evaluations with the full-environment to train an effective 3D airfoil agent—taking a total of 14 hours to train compared to 185 hours for the baseline DRL method.

While this significantly accelerates the time it takes to discover sophisticated controllers for fluid environments, our success also demonstrates the practicality of training agents with

surrogate models using limited computational resources. The SINDy model can easily be evaluated on a single-CPU on a laptop in order to mitigate the real financial costs of running full 3D CFD on HPC systems—effectively lowering the barrier for research groups who can only afford a limited amount of time on HPC clusters.

Finally, we further demonstrate the scalability of our approach to high-dimensional measurement spaces by obtaining observations from 318 sensor measurements from the flow (compared to 70 measurements in the largest 2D example) and projecting into a lower-dimensional space before training. Despite the complex physics, this method is able to capture the dominant behavior of the system needed to learn a controller.

We thank you again for encouraging us to deploy our method on more demanding applications. We believe the addition of the 3D airfoil results significantly strengthens our claims and the relevancy of the paper to the aerospace and fluids community.

SINDy-RL: Interpretable and Efficient Model-Based Reinforcement Learning

Nicholas Zolman, Urban Fasel, J. Nathan Kutz, and Steven L. Brunton

October 21, 2024

The current draft needs improvement. It is difficult to always switch to main draft and supplementary file!

- What do you mean by "interpretable control policy orders of magnitude"?
- I think this sentence should be re-written "In this work, we develop interpretable and generalizable reinforcement learning methods that are also more sample efficient via sparse dictionary learning."
the reason is that "interpretability" means being more specific while "generalization" contradicts it!
- In the Background, this sentence "at each state $u_n \pi(x_n)$ and executing it in the environment, producing a new state x_{n+1} ", should be edited.
- In contributions, what does "Improve sample efficiency by orders of magnitude for training," mean?
- Modify this sentence "By leveraging sparse dictionary learning in tandem with deep reinforcement learning," please do not use "tandem" here, please consider to use usual words.
- Well, it is not always true to say "SINDy" is quite efficient and provide the interpretable models! specially when the size of "library" increases!
One may use another system identification method and fit a model to the given dataset without constructing a big matrix named "library" and use it in practice, like Adaptive control methods. Instead you could say "We tested a new approach, and based on our results performs better than Baseline PPO, etc."
- In Background section, "a more detailed discussion can **be** found in Supplementary,"
- "The agent interacts **with** the environment,"
- r_n (reward, immediate cost, or regret) is not a signal in RL! it is a function!
 $X \times U \rightarrow R$
- Check the related literature in RL/ADP to see how they defined value function/ cost function, etc.

- How do you define the *satisfactory performance* in your algorithm? as you said “We repeat this process and update the dynamics model until the agent has reached satisfactory performance”!
- If you used “D” for “Deep”, then stick to it in the draft — “see deep neural networks” in RL background
- In section 2, SINDy-RL, “Approximating Dynamics,” you say Dyna-style MBRL algorithm! but I could not find it! in the Supplementary file also is not named! you can even use “itemize” in latex to make it better here!
- $y = \dot{x}$, introduce \dot{x} or just say $f(x)$ as you did earlier!
- In equation (1), what is F ! please consider that your reader may not know what is “norm” at all!
- Well, E-SINDy is not necessary robust in the low-data limit!
- Please elaborate on this “Because most dynamic systems (especially with polynomial dynamics) diverge, we bound the state space and reset the surrogate environments during training if a trajectory exits the bounding box.”
- Please elaborate on this sentence “Instead of only sampling points from the surrogate trajectories, we can sample plausible points in a neighborhood surrounding them to create a more accurate approximation while still avoiding regions of the space that can no longer be trusted.”
- You say “x-direction as possible in a fixed time,”! x was your state!
- The word of “surrogate“ it is fuzzy and to tell the truth and hard to digest! you may say one line in introduction about it!
- In the Benchmark section, you introduce several case studies! the reader needs to see in brief their state space x , control inputs u , etc. I saw in the supplementary file “Figure-2“ gives a brief summary of the dynamical systems! so put such figure in the draft! the reason is that reader may not know the dynamic system and simply can have a look at the equations and get the problem!
- One important question that raise is “all the case studies are deterministic, i.e. give a state x_k , and control u_k then there is a unique x_{k+1} ! so why do you say “stochastic policy?” and if so bring an example such as Markov Decision Processes!
- Now lets focus on one case study, e.g “Cartpol swing up”, its state space is defined as $(x, \cos(\theta), \sin(\theta), \dot{x}, \dot{\theta})$, of course we can give values to these and create a large state space! but as we all know in practice engineers do not do such native strategy! this happens in almost in entire industry! such as power plants, motors, etc.

Indeed we say let’s consider a “practical operating point, e.g. $\theta = \frac{\pi}{2}$ then linearize this non-linear model and design a controller for it! this gives a simple, interpret-able, feasible and applicable! or we apply directly non-linear control methods with guaranteed stability!

Now why should I choose an arbitrary policy π_0 and explore “unsafe” state space and meet non-real and non-sense environment?

By the way, why do not you employ “projection based RL approach“ such as Temporal Difference, LSTD, etc.? One can take $(x, \cos(\theta), \sin(\theta), \dot{x}, \dot{\theta})$ as feature vector and project the large state space into the selected features and use Monte Carlo simulations and policy iteration or policy evaluation to compute a vector which can be used as the optimal value/cost function!

why do not you compare the results with conventional methods such as “linear” or “non-linear” or “adaptive” control methods?

- Please include the processor that you used to simulate different case studies.
- Since you use DNN, it is recommended to cite recent works, such as : - “DeepMoD: Deep learning for model discovery in noisy data,” Journal of Computational Physics.
- “A Robust SINDy Approach by Combining Neural Networks and an Integral Form,” <https://arxiv.org/abs/2309.07193>.